# `BetaConform`: Efficient MAP Estimation of LLM Ensemble Judgment Performance with Prior Transfer

**Huaizhi Qu[1], Inyoung Choi[2], Zhen Tan[3], Song Wang[4], Sukwon Yun[1],**
**Qi Long[2], Faizan Siddiqui[5], Kwonjoon Lee[5], Tianlong Chen[1]**
[1]University of North Carolina at Chapel Hill, [2]University of Pennsylvania
[3]Arizona State University [4]University of Virginia [5]Honda Research Institute USA
{huaizhiq, tianlong}@cs.unc.edu

## Abstract

LLM ensembles are widely used for LLM judges. However, how to estimate their accuracy, especially in an efficient way, is unknown. In this paper, we present a principled *maximum a posteriori* (MAP) framework for an economical and precise estimation of the performance of LLM ensemble judgment. We first propose a mixture of Beta-Binomial distributions to model the judgment distribution, revising from the vanilla Binomial distribution. Next, we introduce a conformal prediction-driven approach that enables adaptive stopping during iterative sampling to balance accuracy with efficiency. Furthermore, we design a prior transfer mechanism that utilizes learned distributions on open-source datasets to improve estimation on a target dataset when only scarce annotations are available. Finally, we present `BetaConform`, a framework that integrates our distribution assumption, adaptive stopping, and the prior transfer mechanism to deliver a theoretically guaranteed distribution estimation of LLM ensemble judgment with minimum labeled samples. `BetaConform` is also validated empirically. For instance, with only 10 samples from the TruthfulQA dataset, for a Llama ensembled judge, `BetaConform` gauges its performance with an error margin as small as 3.37%.

## 1 Introduction

With the improving performance of large language models (LLMs), there is a proliferation of adopting LLMs as judges for various tasks [Liang et al., 2023, Yuan et al., 2024b, Zhang et al., 2025]. In applications of LLM judge ensembles, the judgment distribution is critical to the service quality [Chen et al., 2024, Schoenegger et al., 2024, Qiu et al., 2025]. Many datasets [Zheng et al., 2023, Zeng et al., 2023, Yuan et al., 2024a] have been employed to evaluate the performance of LLM judges. However, these datasets rely on human annotations, which are impractical at a large scale due to the substantial time and financial costs of annotating. This challenge highlights the need of *how to estimate the LLM ensemble judging performance efficiently*.

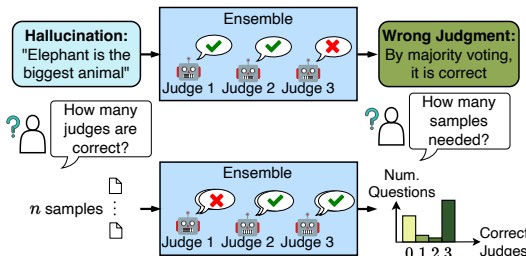

Figure 1: In this paper, we aim to answer (1) how to estimate the judgment distribution of LLM ensemble on a dataset, and (2) how to achieve efficient estimation to reduce annotation effort.

In this work, we consider the following judgment distribution estimation problem:

$$\mathbb{P}(\text{\# correct judgments} = n \mid k \text{ LLMs judge sample } x).$$

We propose an efficient method for MAP estimation of the distribution of LLM ensemble judgment to answer two research questions shown in Figure 1.

39th Conference on Neural Information Processing Systems (NeurIPS 2025).

- **RQ1:** How to efficiently and accurately estimate the judgment distribution?
- **RQ2:** How many samples are needed for estimation under given error margin threshold?

Given a small number of samples, one intuitive estimation is to directly adopt the distribution of the samples as the judgment distribution on the entire dataset. However, this is susceptible to the sampling bias. To avoid this, one common practice is to first calculate the single LLM accuracy on the samples and then model the distribution on the full dataset as Binomial. We first posit that the judgment distribution is not Binomial. Theoretically, a Binomial distribution implies increasing accuracy in majority voting as the ensemble size grows [De Condorcet et al., 2014, Austen-Smith and Banks, 1996]. However, this is unrealistic since the accuracy of LLM ensembles remains bounded even with a large number of judges. To testify to this, we start by observing the distribution of LLM ensemble judges on various benchmarks. We find marked deviations from the Binomial distribution and show a stratification between questions that can be classified as "easy" and "hard". To this end, we propose to model the judgment distribution with a mixture of Beta-Binomial distributions to reflect the stratification. We show that under this assumption, by utilizing an expectation maximization (EM) estimation method, it can achieve accurate judgment distribution estimation with high data efficiency.

To rigorously guide the sampling process and determine how many samples to use for the estimation, we draw inspiration from the conformal prediction (CP) [Shafer and Vovk, 2008, Fontana et al., 2023] that can efficiently estimate the sampling deviation. Based on this, we propose a novel adaptive stopping strategy for iterative sampling, designed to meet a pre-defined deviation threshold. Our experiments demonstrate the effectiveness of this method for limiting the sample amount while maintaining high estimation precision.

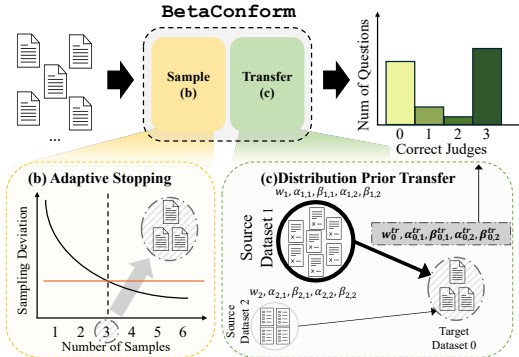

Figure 2: Overview of `BetaConform`. Given a target dataset, adaptive stopping is adopted to determine the sample amount (**b**, Section 5). During iterative sampling, the sampling deviation is monitored by using conformal prediction. The sampling process stops when the deviation is sufficiently low. Next, the estimation of the small number of samples from the previous step is further enhanced by transferring distribution priors from source datasets (**c**, Section 6). The transfer will assign a larger weight to the dataset that is textually closer to the target dataset.

Moreover, we hypothesize that the prior knowledge of judgment distribution on open-source datasets can benefit the estimation of a new dataset when only a few samples are available. To achieve this, we propose a text similarity-based distribution prior transfer mechanism. This method embeds text inputs from both source and target datasets and calculates embedding similarities to determine the transfer weight. Our design greatly improves the estimation accuracy when transferring from similar datasets and avoids performance degradation when the datasets are distinct. Notably, this method relies solely on the text inputs, making it practical for application to vast amounts of unlabeled data.

Our contribution can be summarized as follows:

- We present pioneering work in judgment distribution estimation. We point out that the Binomial assumption of judgment distribution is inaccurate. By replacing it with a mixture of Beta-Binomial distributions, we could achieve efficient and accurate estimation.
- We design a rigorous conformal prediction-based adaptive stopping strategy during iterative sampling when the sampling deviation is sufficiently low.
- We introduce a distribution prior transfer mechanism that leverages judgment distributions on open-source datasets to improve few-sample estimations.
- Extensive experiments show `BetaConform`'s high estimation efficiency. For example, using only 10 samples could result in an average of $10.84\%$ error margin.

## 2   Related Works

**LLMs for Judgment.**   Reliable model evaluation is a critical problem. Traditional human evaluations remain the gold standard, but their scalability is a significant bottleneck in large-scale

applications. Thus, recent works have proposed leveraging LLMs to evaluate the text quality, ranking outputs, and ensuring alignment with human preferences [Zheng et al., 2023, Liu et al., 2023, Dubois et al., 2024]. While initially focused on text generation evaluation, the use of LLMs as judges has expanded to diverse applications including model alignment and safety assessment [Lee et al., 2024], code quality evaluation [Zhao et al., 2024b], and knowledge verification [Min et al., 2023], etc.

**Challenges and Limitations.**   The reliability of such frameworks is not without concerns. Studies have found that even advanced models like GPT-4 often exhibit systematic biases such as position bias and egocentric bias [Zeng et al., 2023, Wang et al., 2023], overconfidence in their judgments [Koo et al., 2024], and self-preference effects [Panickssery et al., 2024]. Moreover, many studies employing LLM annotations do not explicitly measure the alignment between LLMs and humans, thus further raising questions about their dependability [Calderon et al., 2025]. While researchers have proposed various solutions, including dynamic evaluation pipelines [Yu et al., 2024, Zhao et al., 2024a, Moniri et al., 2024], self-reflection mechanisms [Wu et al., 2024, Li et al., 2023b, Wang et al., 2024], and specialized benchmarks for assessing judge performance [Zheng et al., 2023, Tan et al., 2024, Park et al., 2024, Li et al., 2024, Zhao et al., 2024b], these methods often fall short in offering rigorous guarantees of their outcomes. A related line of research is Item Response Theory (IRT) [Cai et al., 2016, Baker, 2001, Harvey and Hammer, 1999], which assesses respondents' latent abilities using responses to calibrated questions. However, the requirement for calibrated questions limits the direct applicability of IRT in the context of judgment distribution estimation, as datasets in this domain are frequently unlabeled.

**Statistical Approaches.**   Another direction of research focuses on providing statistical guarantees for LLM performance. Researchers have explored conformal methods [Angelopoulos et al., 2023] to ensure correctness and factuality [Mohri and Hashimoto, 2024] and to determine when LLMs should abstain from responding [Yadkori et al., 2024]. While these methods provide some statistical rigor, there is still a need for a unified framework that establishes reliable, theoretically grounded approaches for assessing LLM performance across diverse applications.

# 3   Problem Setup

We consider the task of using an LLM ensemble to evaluate and judge samples by discerning, choosing, or scoring. Let:

- $n$: Total number of samples in the dataset to be judged.
- $k$: Number of LLMs in an ensemble.
- $S$: The random variable of correct judgments.
- $r$: Number of samples to estimate $S$.
- $D$: A dataset to estimate the judgment distribution.

**Definition 1** (LLM Ensemble Judgment). Let $\mathcal{J} = \{J_1, J_2, \ldots, J_k\}$ be an ensemble of $k$ LLM judges. For a given input $x$, each LLM $J_i$ generates an output $o_i = J_i(x)$, yielding the set of all judgments $\mathcal{O} = \{o_1, o_2, \ldots, o_k\}$. In this paper, we focus on binary and scoring judgments. We consider the LLM ensemble to be composed of multiple instances of the same underlying model (e.g., $k = 11$ Llama models). Variations in their judgments for a given input are due to $\mathrm{Top-P}$ token sampling [Zhou et al., 2024] and the difference in random seeds.

**Definition 2** (LLM Ensemble Correct Judgment). For an ensemble of $k$ LLMs, the random variable $S = \sum_{i=1}^{k} \mathrm{Match}(o_i, y)$ represents the number of correct judgments. $y$ denotes the ground truth, and $\mathrm{Match}(\cdot)$ is the criterion for a correct judgment. For instance, for binary classification judgments, $\mathrm{Match}(\cdot)$ could be an exact match; for scoring judgments, it could be whether the score falls within a predefined range of the human average score. The ensemble's decision is deemed correct if $S \geq \lceil k/2 \rceil$. *To prevent ties, which can occur if $k$ is an even integer and $S = \lceil k/2 \rceil$, we stipulate that $k$ must be an odd integer.*

# 4   Mixture of Beta-Binomial Distribution

## 4.1   Examination of Binomial Distribution

We start by examining the common assumption of $S$ follows a Binomial distribution, i.e. the probability of having $s$ correct judgments when a single judge accuracy $\hat{p}$ is,

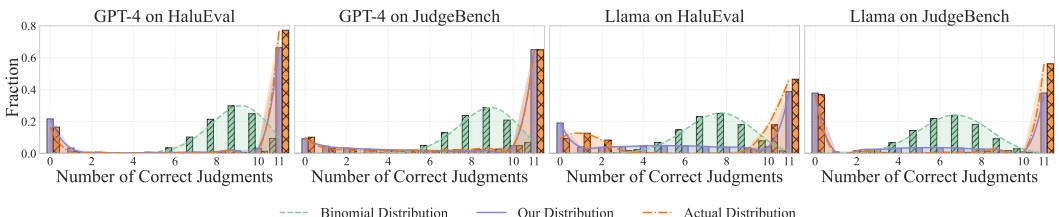

Figure 3: Comparison of judgment distributions among actual, Binomial, and ours. Llama-3.3-70B and GPT-4 ensembles of 11 models are tested on HaluEval and JudgeBench, respectively. The Binomial distribution is estimated by using single judge accuracy $p$. Our mixture distribution is estimated with 100 samples and scaled to the full dataset. **Our distribution is consistently better.**

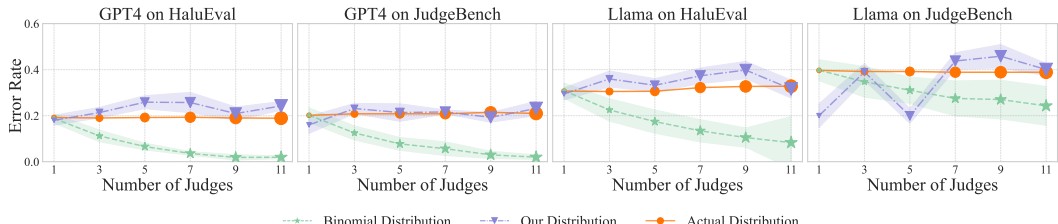

Figure 4: Majority voting error rate of actual, Binomial, and our mixture distribution. Binomial uses single judge accuracy $p$. Our distribution is estimated with 100 random samples and tested for 3 times. The line denotes the average error rate and the shadow represents the standard variance. Binomial shows decreasing error rate, while **our distribution captures the actual trend.**

$$\mathbb{P}_{\text{Bin}}(S = s) = \text{Bin}(s \mid k, \hat{p}) = \binom{k}{s}\hat{p}^s(1 - \hat{p})^{k-s}. \tag{1}$$

The error rate $\tilde{P}_{Bin}$ of ensemble judgment is:

$$\tilde{P}_{\text{Bin}} = \mathbb{P}_{\text{Bin}}(S < \lceil k/2 \rceil) = \sum_{s=0}^{\lceil k/2 \rceil - 1} \binom{k}{s}\hat{p}^s(1 - \hat{p})^{k-s}. \tag{2}$$

We first examine the common assumption that $S$ follows a Binomial distribution in Equation (1). Specifically, we ❶ evaluate individual LLMs on datasets across domains and ❷ use the single LLM accuracy $p$ in Equation (1) and (2) to estimate both the distribution of LLM ensembles on these datasets and the majority voting error rate for different numbers $k$ of LLMs. Specifically, we evaluate GPT-4 [OpenAI et al., 2024] and Llama-3.3-70B [Dubey et al., 2024] on hallucination detection (HaluEval, Li et al., 2023a) and Human alignment (JudgeBench, Tan et al., 2024) datasets. Results are shown in Figure 3 and Figure 4.

The results in Figure 3 and Figure 4 demonstrate the large deviation of Binomial distribution to the real distribution. On both datasets, the real distributions of LLM ensemble judgments consistently show two peaks centering at the two ends, while Binomial distribution results in a single peak with a large shift to either of the two peaks. Notably, in Figure 4, the assumption of a Binomial distribution leads to an always decreasing majority voting error rate, which is in sharp contrast with the actual error rate that remains at the same level when the ensemble becomes larger.

### 4.2 Mixture of Beta-Binomial Distributions

**Assumption 1** (Mixture of Beta-Binomial Distribuitons).
$$S \sim w\text{BB}(k, \alpha_1, \beta_1) + (1 - w)\text{BB}(k, \alpha_2, \beta_2), \tag{3}$$

where $\text{BB}(\cdot, \cdot, \cdot)$ is the Beta-Binomial distribution, $k$ is the number of judges in the ensemble, $\alpha_1, \beta_1, \alpha_2, \beta_2$ are parameters of the two distributions, and $w$ is the mixture weight.

**Corollary 1** (Mixture Distribution Error Rate). The error rate of the mixture of Beta-Binomial distributions is

$$\tilde{P}_{\text{BB}} = w \sum_{s=0}^{\lceil k/2 \rceil - 1} \binom{k}{s}\frac{\text{B}(s + \alpha_1,\ k - s + \beta_1)}{\text{B}(\alpha_1, \beta_1)} + (1 - w) \sum_{s=0}^{\lceil k/2 \rceil - 1} \binom{k}{s}\frac{\text{B}(s + \alpha_2,\ k - s + \beta_2)}{\text{B}(\alpha_2, \beta_2)}, \tag{4}$$

where $B(\cdot, \cdot)$ is the Beta function.

After examining the common Binomial distribution assumption in Figure 3 and Figure 4, we notice that the real distribution keeps showing two peaks centering near all wrong and all correct. Motivated by this observation, in Assumption 1 we model the distribution as a mixture of two Beta-Binomial distributions, where one distribution models the LLM ensemble judgments on simple questions and the other one for hard problems. To derive all the parameters, we utilize labeled samples from the dataset and design a distribution-tailored expectation maximization (EM) algorithm.

### 4.3 Expectation Maximization

**Samples as Distribution Evidence.** Given $r$ samples, each containing judgments from $k$ LLMs, $S_i$ is the number of correct judgments in the $i$-th sample and $p_i = S_i/k$ as the estimated probability of success for the $i$-th sample.

For the $i$-th sample, considering the first Beta-Binomial distribution, a responsibility $\gamma_1^i$ is assigned as

$$\gamma_1^i = \frac{w\text{Beta}(p_i \mid \alpha_1, \beta_1)}{w\text{Beta}(p_i \mid \alpha_1, \beta_1) + (1 - w)\text{Beta}(p_i \mid \alpha_2, \beta_2)}, \tag{5}$$

where $\text{Beta}(p_i \mid \alpha, \beta)$ is the probability density of beta distribution at $p_i$ for the $i$-th sample under the corresponding Beta component. $\gamma_1^i$ represents the probability that the $i$-th sample belongs to the first Beta component, and $\gamma_2^i = 1 - \gamma_1^i$ is the probability for the second component.

**Parameters Update.** The parameters are updated based on the weighted contributions of samples. The parameters of two distributions $j = \{1, 2\}$ are updated as

$$\alpha_j' = \sum_{i=1}^{r} \gamma_1^i \cdot S_i, \ \beta_j' = \sum_{i=1}^{r} \gamma_1^i \cdot (k - S_i), \ w' = \frac{1}{r}\sum_{i=1}^{r} \gamma_1^i \tag{6}$$

We verify our distribution assumption by first sampling $r = 100$ judgments made by two models on two datasets and apply our distribution-tailored EM algorithm to estimate the parameters. Our method is evaluated in two scenarios: ❶ In Figure 3, we fix the ensemble size $k = 11$ and compare the estimated distribution against the real distribution and Binomial distribution, and ❷ in Figure 4 we estimate the error rate of majority voting with different ensemble sizes.

In Figure 3, the mixture of Beta-Binomial distributions is significantly closer to the real distribution compared to the Binomial, with clear two-peak patterns that are analogous to the observation. In Figure 4 it shows that our distribution is consistently close to the real majority voting error rate across all ensemble sizes. Contrary to the Binomial distribution that produced a decreasing error rate, our distribution successfully modeled the stable error rate when the ensemble becomes larger. Additionally, the narrow confidence interval demonstrates the high stability of our method.

## 5 Guide Sampling via Conformal Prediction

In the experiments above, we used a fixed number of samples. However, in practical settings where datasets are unannotated and being labeled, it is essential to determine when the number of annotated samples is sufficient for accurate estimation. Inspired by conformal prediction (CP), which does not rely on prior knowledge of the dataset distribution and can rigorously estimate the sampling deviation, we propose leveraging its principles to address this challenge.

### 5.1 Conformal Prediction for Adaptive Stopping

CP provides a principled approach to dynamically evaluate the sampling deviation in the distribution of the number of correct judgments $S$, which can be used as guidance.

**Nonconformity Scores.** A major part of CP is the nonconformity score, which measures how a test sample differs from the rest of the data. In our implementation, we set the nonconformity score as

$$\text{score}(S_i) = |S_i - \mathbb{E}[S]|, \tag{7}$$

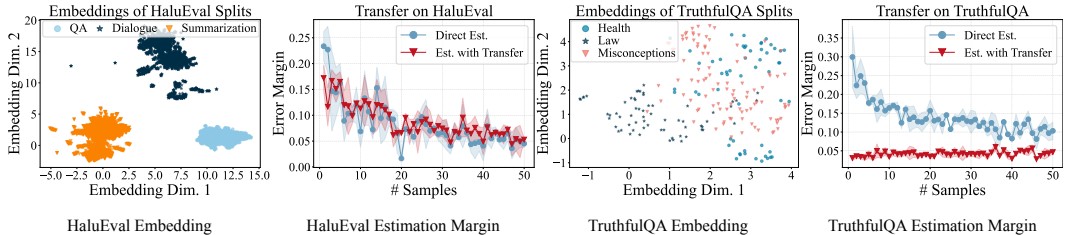

| HaluEval Embedding | HaluEval Estimation Margin | TruthfulQA Embedding | TruthfulQA Estimation Margin |

Figure 5: Examples of distribution prior transfer. Splits from HaluEval form distinct clusters in the embedding space, and transfer does not degrade performance compared to only using target dataset samples. In contrast, topics in TruthfulQA exhibit closer proximity, where transfer leads to significant performance improvements compared to solely using the limited samples of the target dataset.

which quantifies the deviation of each observed value of $S$ from the expected value.

**Calibration Data and Quantile Computation.** Suppose $r$ samples have been used to test the LLM ensemble with $S_1, S_2, \ldots, S_r$ correct judgments, the CP sampling computes the nonconformity scores for all calibration data as $s_i = \text{score}(S_i)$ and these scores are sorted in ascending order as $s_1 < \ldots < s_r$. For a desired estimation confidence $1 - \epsilon$, the $(1 - \epsilon)$-quantile with $r$ samples $q_{1-\epsilon}^r$ is

$$q_{1-\epsilon}^r = s_{\lceil (1-\epsilon) \cdot (r+1) \rceil}. \tag{8}$$

**Adaptive Stopping Criteria.** Adaptive stopping is achieved by monitoring the variation of the conformal prediction quantile. After $r$ samples, the $(1 - \epsilon)$-quantile is recomputed and compared with the one from $r - 1$ samples. The sampling process stops when the quantile satisfies

$$\left| q_{1-\epsilon}^r - q_{1-\epsilon}^{r-1} \right| \leq \xi \tag{9}$$

where $\xi$ is a predefined threshold.

**Proposition 1** (Sample Amount with Adaptive Stopping). For a given sampling deviation threshold $\xi$ and a scale $\tau$, the sample amount $r$ should satisfy

$$\tau \left( \frac{1}{\sqrt{r-1}} - \frac{1}{\sqrt{r}} \right) \leq \xi, \tag{10}$$

This proposition offers an estimation of the sample amount under the threshold $\xi$.

**Proposition 2** (Error Rate with Adaptive Stopping). Under the sampling threshold $\xi$, the majority voting error rate of the mixture distribution becomes

$$(1 - \min(\xi, \frac{\tau}{\sqrt{r}})) \tilde{P}_{\text{BB}} < \tilde{P}_{\text{adapt}} < (1 + \min(\xi, \frac{\tau}{\sqrt{r}})) \tilde{P}_{\text{BB}} \tag{11}$$

This proposition provides a theoretical error bound for estimation under adaptive stopping, suggesting the mild degradation of estimation performance.

We leave the proofs of Proposition 1 and 2 in Appendix B.1 and B.2, respectively. In our experiments, we set $\xi = 0.03$, and $\tau = 25$, which leads to $r \geq 56$.

## 6 Text Similarity for Distribution Prior Transfer

To further improve the data efficiency when only a few samples are available and enhance estimation accuracy, we propose to incorporate prior knowledge about the LLM ensemble on other open-source datasets and transfer the estimated judgment distributions to the target dataset. However, one challenge is that the prior transfer could bring performance degradation if the distributions of the source datasets and the target dataset are very different. To resolve this challenge, we design text similarity-based distribution prior transfer, which leverages the strong text embedding capability of the recent models to understand and measure the textual difference among datasets.

**Text Embedding.** To embed the text inputs of the LLM ensemble, we use NV-Embed-V2 [Lee et al., 2025]. Given sets of samples $\{D_1, D_2, \ldots, D_m\}$ from $m$ source datasets, the embedding model $\mathcal{E}(\cdot)$ is utilized to transform the sets of samples to sets of embeddings for the source datasets

$$\{E_1, E_2, \ldots, E_m\} = \{\mathcal{E}(D_1), \mathcal{E}(D_2), \ldots, \mathcal{E}(D_m)\}. \tag{12}$$

The average embedding $\bar{E}_i = \frac{1}{r_i} \sum_{j=1}^{r_i} E_i^j$ of the $i$-th dataset is used to represent it.

**Distribution Prior Transfer.** To transfer the distribution from source datasets to the target dataset $D_0$, the process starts by embedding the target dataset $E_0 = \mathcal{E}(D_0)$ and acquiring its average embedding $\bar{E}_0$. For the dataset $D_i$, its transfer weight is

$$\lambda_i = \log(r_i) \cdot \sigma\left(\rho_1 \cdot \left(\text{CosSim}\left(\bar{E}_0, \bar{E}_i\right) - \rho_2\right)\right), \tag{13}$$

where $\sigma(\cdot)$ is the sigmoid function, $r_i$ is the number of samples and $\rho_1$ and $\rho_2$ are hyperparameters. We adopt this design to avoid the degradation of estimation caused by transferring datasets with dissimilar text inputs. This is achieved by setting a threshold and applying the sigmoid function to suppress the weight when the similarity is low. $\log(r_i)$ is included as datasets with more samples could produce a more accurate estimation and thus should have a higher impact on the transfer. The transfer from the source datasets to the target dataset is performed as

$$w_0^{tr} = \frac{\sum_{i=0}^m \lambda_i \cdot w_i}{\sum_{i=0}^m \lambda_i}, \alpha_{0,j}^{tr} = \frac{\sum_{i=0}^m \lambda_i \cdot \alpha_{i,j}}{\sum_{i=0}^m \lambda_i}, \beta_{0,j}^{tr} = \frac{\sum_{i=0}^m \lambda_i \cdot \beta_{i,j}}{\sum_{i=0}^m \lambda_i}, \quad j \in \{1, 2\}. \tag{14}$$

In Equation (14), $\alpha_{i,j}$ and $\beta_{i,j}$ are the $j$-th parameter in the mixture distribution of $i$-th dataset. The parameters in the weighted sum with index 0 denote direct estimation on the target dataset.

**Examples.** To verify our distribution design, we evaluate the distribution within splits of HaluEval [Li et al., 2023a] and TruthfulQA [Lin et al., 2021] datasets. For HaluEval, we use Dialogue and Summarization splits as source datasets and transfer to QA split; for TruthfulQA, we transfer from topics of Health and Law to Misconceptions. As shown in Figure 5, the embeddings form distant clusters in HaluEval, as the text inputs of the three splits have different hallucination detection requirements, and embeddings from TruthfulQA overlap due to the similarity of judgment format. When clusters are separated, our method will not bring performance degradation compared to solely using samples from the target dataset, while when clusters are overlapping, our method brings a significantly lower estimation error rate margin compared to only using target dataset samples. This supports the effectiveness of our distribution transfer design.

We present the algorithm and Python implementation of `BetaConform` in Section A and Section E.

# 7 Experiments

## 7.1 Estimation Accuracy

We begin by evaluating `BetaConform` with adaptive stopping on datasets to verify its accuracy. We choose Binomial distribution and a single Beta-Binomial distribution as baselines and compare the error margin, which is the absolute difference between the estimation error rate and the actual value. The results are reported in Table 1. Please see Section E for implementation details.

From the results, the following observations can be drawn: ❶ Compared to the Binomial distribution, `BetaConform` achieves consistently lower error margin, with $32.4\% \sim 54.1\%$ improvements of average error margin of all models. This demonstrates an effective answer to **RQ1** by modeling judgment distribution as a mixture of Beta-Binomial distributions. ❷ The number of samples is close to the theoretical estimation. The average sample amount of models on all datasets exhibit a slight deviation of the estimated value 56 by $3.14 \sim 12.86$ samples. This validates our design of using the distribution-free CP for adaptive stopping, which effectively solved **RQ2**.

## 7.2 Distribution Prior Transfer

We then verify our text similarity-based distribution prior transfer when only limited samples are available. We constrain to 10 samples from the target dataset and assume the full source datasets are accessible. Transfer is compared with estimating only on the target dataset samples (w/o Transfer). Error margins are shown in Table 2. We also conduct ablation studies of the transfer design in Table 4

Table 1: The comparison of error margins between our mixture of Beta-Binomial distributions and Binomial distribution. The **Err. Margin** and **# Samples** answer **RQ1** and **RQ2**, respectively. The error margin is calculated as the absolute difference between the actual error rate and the estimation. Estimations using both distributions are done on samples obtained through iterative sampling with adaptive stopping. For each run, the error margin is computed from $k = 1$ to $11$, and the average margin of ensemble sizes is used as the result for that run. We conduct $30$ runs and report the average and standard deviation. The average number of samples across runs is also reported.

| Dataset | Method | Llama-3.3-70B Error Margin (↓) | # Samples (↓) | Qwen-2.5-72B Error Margin (↓) | # Samples (↓) | InternLM-20B Error Margin (↓) | # Samples (↓) | GPT-3.5 Error Margin (↓) | # Samples (↓) | GPT-4 Error Margin (↓) | # Samples (↓) |
|---|---|---|---|---|---|---|---|---|---|---|---|
| *Hallucination Detection* | | | | | | | | | | | |
| HaluEval | Binomial | 17.62 ± 0.73 | | 12.45 ± 1.04 | | 16.67 ± 0.38 | | 5.78 ± 0.08 | | 9.16 ± 0.18 | |
| | Single BB | 14.46 ± 0.16 | 49.47 | 5.14 ± 0.21 | 61.02 | 15.92 ± 0.11 | 50.67 | 5.27 ± 0.09 | 34.80 | 9.77 ± 0.84 | 40.58 |
| | Ours | **6.68 ± 0.53** | | **4.72 ± 0.38** | | **5.48 ± 0.41** | | **5.10 ± 0.24** | | **6.28 ± 0.39** | |
| TruthfulQA | Binomial | 14.00 ± 0.65 | | 19.86 ± 0.40 | | 19.55 ± 0.65 | | 14.44 ± 0.40 | | 15.20 ± 0.55 | |
| | Single BB | 8.83 ± 1.02 | 54.13 | 7.84 ± 0.26 | 53.56 | 6.79 ± 0.25 | 55.56 | 12.17 ± 0.99 | 47.64 | 11.31 ± 0.52 | 57.07 |
| | Ours | **7.53 ± 0.55** | | **7.18 ± 0.44** | | **6.24 ± 0.59** | | **6.75 ± 0.58** | | **6.73 ± 0.38** | |
| HalluDial | Binomial | 13.10 ± 0.37 | | 13.42 ± 0.54 | | 14.84 ± 0.42 | | 8.79 ± 0.21 | | 9.25 ± 0.27 | |
| | Single BB | 11.33 ± 0.64 | 46.58 | 16.75 ± 0.90 | 55.78 | 7.95 ± 0.34 | 51.87 | 9.24 ± 0.59 | 41.51 | 8.43 ± 0.45 | 42.31 |
| | Ours | **7.94 ± 0.68** | | **6.96 ± 0.47** | | **6.43 ± 0.50** | | **6.27 ± 0.36** | | **5.22 ± 0.59** | |
| *Reasoning* | | | | | | | | | | | |
| PRM800K | Binomial | 10.11 ± 0.29 | | 9.14 ± 0.17 | | 9.12 ± 0.20 | | 8.83 ± 0.25 | | 14.52 ± 0.73 | |
| | Single BB | 16.45 ± 1.35 | 43.33 | 10.30 ± 0.60 | 42.89 | 9.81 ± 0.61 | 46.13 | 9.45 ± 0.72 | 51.38 | 12.31 ± 0.31 | 54.67 |
| | Ours | **9.37 ± 0.64** | | **7.82 ± 0.69** | | **4.52 ± 0.50** | | **8.46 ± 0.51** | | **6.17 ± 0.48** | |
| BIG-bench | Binomial | 13.29 ± 0.78 | | 14.17 ± 0.40 | | 14.68 ± 0.24 | | 14.83 ± 0.53 | | 12.15 ± 0.74 | |
| | Single BB | 13.15 ± 0.68 | 51.51 | 12.32 ± 0.60 | 47.82 | 9.51 ± 0.56 | 48.40 | 17.93 ± 0.89 | 46.13 | 11.50 ± 0.91 | 46.09 |
| | Ours | **11.15 ± 0.60** | | **6.97 ± 0.58** | | **5.54 ± 0.51** | | **12.59 ± 0.48** | | **8.02 ± 0.59** | |
| TRAM | Binomial | 14.79 ± 0.82 | | 13.13 ± 0.64 | | 13.06 ± 0.77 | | 4.99 ± 0.13 | | 5.14 ± 0.11 | |
| | Single BB | 11.75 ± 0.74 | 55.87 | **5.72 ± 0.39** | 57.16 | **6.01 ± 0.44** | 57.78 | 7.42 ± 0.14 | 39.07 | **4.01 ± 0.30** | 38.53 |
| | Ours | **8.39 ± 0.63** | | 6.20 ± 0.34 | | 6.10 ± 0.58 | | **3.94 ± 0.17** | | 4.81 ± 0.23 | |
| *Alignment* | | | | | | | | | | | |
| JudgeBench | Binomial | 12.06 ± 0.78 | | 13.45 ± 0.54 | | 10.31 ± 1.03 | | 8.85 ± 0.33 | | 10.98 ± 0.32 | |
| | Single BB | 7.60 ± 0.37 | 60.58 | 7.64 ± 0.54 | 58.40 | **5.11 ± 0.24** | 57.16 | 11.85 ± 0.78 | 41.07 | 7.62 ± 0.25 | 46.58 |
| | Ours | **6.98 ± 0.56** | | **5.39 ± 0.39** | | 5.26 ± 0.39 | | **7.03 ± 0.61** | | **6.45 ± 0.53** | |
| RewardBench | Binomial | 8.40 ± 0.19 | | 8.93 ± 0.22 | | 17.36 ± 1.41 | | 11.42 ± 0.33 | | 13.98 ± 0.29 | |
| | Single BB | 16.29 ± 1.39 | 40.22 | 11.40 ± 1.20 | 45.20 | **6.15 ± 0.27** | 52.04 | 8.79 ± 0.21 | 42.27 | 8.80 ± 0.40 | 48.22 |
| | Ours | **11.30 ± 0.62** | | **4.68 ± 0.56** | | 6.58 ± 0.40 | | **6.90 ± 0.45** | | **7.65 ± 0.51** | |
| LLMBar | Binomial | 13.61 ± 0.58 | | 14.63 ± 0.51 | | 13.66 ± 1.14 | | **13.19 ± 0.55** | | 10.36 ± 0.33 | |
| | Single BB | 14.21 ± 0.67 | 50.18 | 7.97 ± 0.58 | 51.07 | **5.46 ± 0.30** | 51.29 | 13.46 ± 0.83 | 44.40 | 11.72 ± 0.48 | 44.40 |
| | Ours | **10.18 ± 0.71** | | **7.52 ± 0.63** | | 6.38 ± 0.53 | | 13.71 ± 0.54 | | **8.16 ± 0.50** | |
| *Scoring* | | | | | | | | | | | |
| ICE-Score | Binomial | **8.91 ± 0.25** | | 9.27 ± 0.23 | | 22.24 ± 1.02 | | 3.61 ± 0.06 | | **3.66 ± 0.07** | |
| | Single BB | 16.71 ± 1.11 | 41.29 | 9.24 ± 0.59 | 43.73 | **10.97 ± 0.27** | 53.42 | 3.54 ± 0.22 | 39.87 | 4.69 ± 0.10 | 38.93 |
| | Ours | 8.97 ± 0.45 | | **6.91 ± 0.59** | | 18.19 ± 0.37 | | **3.39 ± 0.32** | | 5.78 ± 0.08 | |
| COMP-Analysis | Binomial | 14.45 ± 0.71 | | 15.88 ± 0.72 | | 13.28 ± 0.73 | | 12.87 ± 0.32 | | 15.64 ± 0.68 | |
| | Single BB | 8.56 ± 0.66 | 53.91 | **6.93 ± 0.34** | 53.33 | **4.61 ± 0.27** | 57.11 | 7.82 ± 0.29 | 46.40 | 11.32 ± 0.43 | 53.82 |
| | Ours | **6.50 ± 0.63** | | 6.95 ± 0.50 | | 4.86 ± 0.48 | | **6.66 ± 0.38** | | **7.07 ± 0.48** | |
| *Average* | | | | | | | | | | | |
| Average | Binomial | 12.76 ± 0.56 | | 13.12 ± 0.49 | | 14.98 ± 0.73 | | 9.78 ± 0.29 | | 10.91 ± 0.39 | |
| | Single BB | 12.67 ± 0.61 | 49.73 | 9.20 ± 0.56 | 51.81 | 8.03 ± 0.33 | 52.86 | 9.72 ± 0.52 | 43.14 | 10.11 ± 0.45 | 46.47 |
| | Ours | **8.63 ± 0.60** | | **6.48 ± 0.51** | | **6.87 ± 0.48** | | **7.35 ± 0.42** | | **6.38 ± 0.44** | |

Figure 6: The actual number of samples under various thresholds $\xi$ versus the theoretical value from Equation (10). **The actual sample numbers match with the theoretical bound.**

Figure 7: The actual number of samples from different datasets under three $\xi$ values. **Our sampling with adaptive stopping shows consistent results on all the datasets.**

From the results, we observe that by transferring from other datasets in the same category (e.g., from TruthfulQA and HalluDial to HaluEval), the average error margin across all datasets is reduced by $5.0\% \sim 25.0\%$ and is consistently lower compared to no transfer, suggesting the effectiveness of using prior knowledge of the judgment distributions on open-source datasets can benefit estimation.

### 7.3 More Research Questions

**RQ3: Is sampling with adaptive stopping consistent to the theory?** We examine our adaptive stopping to see if Equation (10) matches the real sampling amount. We set a series of $\xi$ while keeping $\tau = 25$ and sample with adaptive stopping from judgment samples produced by Llama, Qwen, and GPT-4, and compare with the theoretical value of Equation (10). The actual sample amounts under different thresholds in Figure 6 match closely with the theoretical estimation, which proves the effectiveness of quantifying sampling deviation through CP and the Proposition 1.

**RQ4: Is adaptive stopping really distribution-free?** One benefit of adopting CP to quantify sampling deviation is distribution irrelevance. To testify to this, we consider sampling with various thresholds on all datasets to see if the sample amount remains consistent. The results in Figure 7

Table 2: The comparison of error margins with and without distribution prior transfer. Estimations are performed using the mixture of Beta-Binomial distributions, with 10 samples randomly drawn for estimation. In experiments, each dataset is chosen as the target dataset, and the left datasets in the same domain are used as source datasets. **Bold** denotes lower margin. Scores are in percent (%).

| Dataset | Method | Llama-3.3-70B | Qwen-2.5-72B | InternLM-2.5-20B | GPT-3.5 | GPT-4 |
|---|---|---|---|---|---|---|
| | | | Hallucination Detection Datasets | | | |
| HaluEval | w/o Transfer | 12.43 ± 0.87 | 12.50 ± 0.92 | 10.09 ± 0.64 | 14.07 ± 0.75 | 12.85 ± 0.83 |
| | w/ Transfer | **8.82 ± 0.42** | **9.19 ± 0.75** | **8.60 ± 0.64** | **8.88 ± 0.71** | **8.88 ± 0.86** |
| TruthfulQA | w/o Transfer | 15.30 ± 0.81 | 13.88 ± 0.85 | 13.17 ± 1.11 | 12.54 ± 0.70 | 13.21 ± 1.03 |
| | w/ Transfer | **3.37 ± 0.10** | **8.55 ± 0.07** | **10.18 ± 0.10** | **10.18 ± 0.82** | **9.66 ± 0.70** |
| HalluDial | w/o Transfer | 17.53 ± 0.81 | 16.15 ± 0.60 | 11.35 ± 0.83 | **16.62 ± 0.70** | **14.64 ± 0.85** |
| | w/ Transfer | **12.89 ± 0.77** | **13.42 ± 0.53** | **8.72 ± 0.54** | 23.79 ± 0.84 | 18.77 ± 0.92 |
| | | | Reasoning Datasets | | | |
| PRM800K | w/o Transfer | **15.02 ± 0.78** | 12.85 ± 0.88 | **8.22 ± 0.58** | **9.27 ± 0.84** | 9.97 ± 0.53 |
| | w/ Transfer | 15.11 ± 0.62 | **10.96 ± 0.99** | 8.46 ± 0.60 | 10.55 ± 0.84 | **9.71 ± 1.00** |
| BIG-bench | w/o Transfer | 15.22 ± 0.74 | **13.81 ± 0.82** | **9.44 ± 0.53** | 14.39 ± 0.74 | 13.31 ± 1.15 |
| | w/ Transfer | **12.69 ± 0.74** | 14.28 ± 0.79 | 10.00 ± 0.62 | **9.98 ± 0.67** | **13.22 ± 0.69** |
| TRAM | w/o Transfer | 14.77 ± 0.84 | 12.27 ± 0.69 | 11.67 ± 0.76 | **13.52 ± 0.81** | **12.69 ± 1.26** |
| | w/ Transfer | **12.52 ± 0.92** | **11.03 ± 1.04** | **10.85 ± 0.97** | 11.81 ± 1.00 | 11.25 ± 0.57 |
| | | | Alignment Datasets | | | |
| JudgeBench | w/o Transfer | 14.05 ± 0.88 | 12.41 ± 0.66 | 11.37 ± 0.79 | **8.23 ± 0.75** | **12.32 ± 0.69** |
| | w/ Transfer | **9.45 ± 0.59** | **8.19 ± 0.66** | **8.03 ± 0.54** | 14.36 ± 0.68 | 15.30 ± 1.19 |
| RewardBench | w/o Transfer | 12.73 ± 0.68 | **9.47 ± 1.07** | **10.34 ± 0.67** | **15.17 ± 0.92** | 13.30 ± 0.77 |
| | w/ Transfer | **12.72 ± 0.30** | 12.84 ± 0.48 | 16.35 ± 0.36 | 18.12 ± 0.34 | **12.57 ± 0.38** |
| LLMBar | w/o Transfer | 16.97 ± 1.10 | 15.91 ± 0.70 | 10.03 ± 0.88 | **17.00 ± 0.64** | **12.90 ± 0.97** |
| | w/ Transfer | **8.03 ± 0.39** | **9.95 ± 0.30** | **8.61 ± 0.41** | 21.94 ± 0.42 | 17.70 ± 0.40 |
| | | | Scoring Datasets | | | |
| ICE-Score | w/o Transfer | 14.08 ± 0.53 | **11.90 ± 1.05** | 19.59 ± 0.78 | 12.11 ± 0.82 | 13.98 ± 0.88 |
| | w/ Transfer | **11.32 ± 0.66** | 11.99 ± 0.76 | **19.25 ± 1.05** | **10.63 ± 0.66** | **12.30 ± 0.67** |
| COMP-Analysis | w/o Transfer | **14.85 ± 1.45** | **10.83 ± 0.60** | 10.29 ± 0.60 | 10.22 ± 0.53 | 16.18 ± 1.00 |
| | w/ Transfer | 15.29 ± 0.91 | 12.28 ± 1.38 | **10.23 ± 0.72** | **9.62 ± 0.53** | **14.97 ± 0.82** |
| | | | Average | | | |
| Average | w/o Transfer | 14.81 ± 0.86 | 12.91 ± 0.80 | 11.41 ± 0.74 | **13.01 ± 0.75** | 13.21 ± 0.91 |
| | w/ Transfer | **11.11 ± 0.58** | **11.15 ± 0.70** | **10.84 ± 0.60** | 13.62 ± 0.68 | **13.12 ± 0.74** |

show only a slight variance of sampling amounts across datasets, demonstrating superior stability. This verifies that our adaptive stopping is truly distribution-free, and stable on diverse datasets.

**RQ5: Is CP-based Adaptive Stopping efficient?**   To validate the effectiveness of our CP-based adaptive stopping, we compare it against variance-based stopping. Specifically, we calculate the variance of sampling as

$$\text{Var}(sampling) = \frac{\alpha_r \beta_r}{(\alpha_r + \beta_r)^2 (\alpha_r + \beta_r + 1)}, \tag{15}$$

where $\alpha_r$ and $\beta_r = r - \alpha_r$ are the number of correct and wrong judgments in $r$ samples, respectively.

As shown in Table 3, is consistently more effective for adaptive stopping under the same deviation threshold $\xi$, which results in a reduced number of samples and achieves a reduction of up to 46.3%.

## 8  Conclusion

We present `BetaConform`, a framework for efficient estimation of LLM ensemble judge distribution. As part of our framework, we propose a mixture of Beta-Binomial distributions to model the judgment distribution after examining the inaccuracy of the Binomial assumption. We design conformal prediction-based adaptive stopping for sampling, which monitors the sampling deviation and effectively determines the sample amount for estimation. When only limited samples are available, we incorporate a text similarity-based distribution prior transfer mechanism to improve the estimation accuracy. As shown by experiments, the conformal prediction-based adaptive stopping effectively guided the sampling. Our mixture of Beta-Binomial distributions significantly outperforms the common Binomial assumption. With the transfer mechanism, `BetaConform` can achieve high estimation precision with as few as 10 samples from the target dataset.

Table 3: Comparison of variance-based adaptive stopping and ours. We compare the sample amount of both methods under the same threshold. **Bold** denotes less samples.

| Threshold $\xi$ | Methods | HaluEval | JudgeBench | PRM800K | ICE-Score |
|---|---|---|---|---|---|
| | | # Samples (↓) | # Samples (↓) | # Samples (↓) | # Samples (↓) |
| $\xi=0.06$ | Variance | 36.87 | 36.87 | 26.00 | **24.77** |
| | Ours | **35.37** | **36.37** | **30.47** | 31.53 |
| $\xi=0.03$ | Variance | 82.09 | 74.43 | 79.76 | 81.47 |
| | Ours | **54.72** | **53.90** | **43.32** | **45.27** |
| $\xi=0.01$ | Variance | 194.72 | 198.56 | 147.22 | 151.44 |
| | Ours | **109.06** | **106.56** | **101.28** | **96.50** |

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

## A  BetaConform

In this section, we introduce BetaConform, a framework designed for the efficient estimation of judgment distributions, as illustrated in Figure 2 and Algorithm 1. The framework operates in two scenarios: when only limited samples are available on the target dataset, and when a larger number of samples can be collected. In the former case, BetaConform leverages prior distributions from source datasets to enhance estimation. In the latter, it employs adaptive stopping during iterative sampling to balance sample efficiency and estimation accuracy.

(1) When only a small number of samples are available from the target dataset, BetaConform follows these steps: ❶ First, it estimates the mixture of Beta-Binomial distributions using the available samples. ❷ Next, it incorporates prior knowledge by transferring distributions from source datasets. Specifically, it estimates the distributions on the source datasets using all available samples and calculates transfer weights based on Equation 13. ❸ Finally, the distributions from the source datasets are aggregated using Equation 14 to produce an enhanced estimation for the target dataset.

---

**Algorithm 1 BetaConform**

1: **Input:** target dataset $D_0$, source datasets $D_1, \ldots, D_m$, judges $\mathcal{J} = \{J_1, \ldots, J_k\}$, EM algorithm $\mathrm{EM}(\cdot)$
2: **Output:** distribution parameters $\Omega$ on the target dataset
3: **if** limited samples in $D_0$ **then**
4:     Compute distribution parameters on $D_0$
5:     Compute parameters of distributions on $D_1, \ldots, D_m$
6:     Compute transfer weights by Equation (13)
7:     $\Omega \leftarrow$ Compute transferred parameters by Eq. (14)
8: **else**
9:     Initial $D \leftarrow \{\}$, $q_{1-\epsilon}^0 \leftarrow -\infty$
10:     **while** Equation (9) is not satisfied **do**
11:         Add a sample from $D_0$ to $D$ and update $q_{1-\epsilon}^{|D|}$
12:     **end while**
13:     $\Omega \leftarrow$ Compute distribution parameters on samples $D$
14: **end if**
15: **return** $\Omega$

---

(2) When the target dataset contains a large number of unlabeled samples, BetaConform employs the following process: ❶ It uses a conformal prediction (CP)-based adaptive stopping strategy to guide the labeling process. ❷ During iterative sampling, batches of samples are drawn and labeled, while the variation in the nonconformity score is monitored. The sampling process stops when the variation falls below a predefined threshold. ❸ Once sufficient labeled samples are collected, the mixture of Beta-Binomial distributions is directly estimated using these samples.

## B  Proofs

### B.1  Determination of Sample Amount.

To derive a theoretical estimation of the sample amount for the adaptive stopping criteria above, we utilize the fundamental statistical properties of variance reduction with increasing sample size. Specifically, for i.i.d samples, the variance of the quantile decreases as:

$$\mathrm{Var}(q_{1-\epsilon}^r) \propto \frac{1}{r \cdot f(q_{1-\epsilon})^2}, \tag{16}$$

where $f(q_{1-\epsilon})$ is the density function at the quantile. The standard deviation of the estimator, which determines the variability of the quantile estimate, thus decays as:

$$\mathrm{StdDev}(q_{1-\epsilon}^r) \propto \frac{1}{\sqrt{r}}. \tag{17}$$

By the asymptotic theory of quantile estimation, for a large enough number of samples $r$, the empirical quantile $q_{1-\epsilon}^r$ converges to the quantile on the whole dataset $q_{1-\epsilon}$ with a known distribution based on Bahadur's representation:

$$\sqrt{r}\left(q_{1-\epsilon}^r - q_{1-\epsilon}\right) \sim \mathcal{N}\left(0, \frac{\epsilon(1-\epsilon)}{f(q_{1-\epsilon})^2}\right), \tag{18}$$

This implies:

$$q_{1-\epsilon}^r = q_{1-\epsilon} + O_p\left(\frac{1}{\sqrt{r}}\right), \tag{19}$$

where $O_p(\cdot)$ denotes the order in probability. Thus, we can determine that the quantile itself decays as:

$$q_{1-\epsilon}^r - q_{1-\epsilon} = O_p\left(\frac{1}{\sqrt{r}}\right). \tag{20}$$

This decay behavior shows that as $r$ increases, the estimated quantile approaches the theoretical quantile $q_{1-\epsilon}$, reflecting decreasing sampling deviation by using more samples. We will use this property to derive the relationship between the stopping criteria and the sample size $r$. From the stopping criteria in Equation (9),

$$\left|q_{1-\epsilon}^r - q_{1-\epsilon}^{r-1}\right| \leq \xi. \tag{21}$$

According to the calculations in Equation (20), we can rewrite the bound for $q_{1-\epsilon}^{r-1}$ as

$$q_{1-\epsilon}^{r-1} - q_{1-\epsilon} = O_p\left(\frac{1}{\sqrt{r-1}}\right). \tag{22}$$

Thus we have

$$\left|q_{1-\epsilon}^r - q_{1-\epsilon}^{r-1}\right| = O_p\left(\frac{1}{\sqrt{r}} - \frac{1}{\sqrt{r-1}}\right). \tag{23}$$

This suggests to meet Equation (9), it requires

$$\tau\left(\frac{1}{\sqrt{r-1}} - \frac{1}{\sqrt{r}}\right) < \xi, \tag{24}$$

which proves Equation (10).

## B.2 Error Rate with Adaptive Sampling

In this section we develop a theoretical estimation of the error bound for adaptive sampling. We first consider the base case and as shown in Equation (4), we know that the mixture distribution error rate is:

$$\tilde{P}_{\text{BB}} = w \sum_{s=0}^{\lceil k/2 \rceil - 1} \binom{k}{s} \frac{\text{B}(s + \alpha_1, k - s + \beta_1)}{\text{B}(\alpha_1, \beta_1)} + (1-w) \sum_{s=0}^{\lceil k/2 \rceil - 1} \binom{k}{s} \frac{\text{B}(s + \alpha_2, k - s + \beta_2)}{\text{B}(\alpha_2, \beta_2)} \tag{25}$$

The adaptive stopping criterion is given by Equation (9):

$$\left|q_{1-\epsilon}^r - q_{1-\epsilon}^{r-1}\right| \leq \xi. \tag{26}$$

The sample size requirement is given by Equation (10):

$$\tau\left(\frac{1}{\sqrt{r-1}} - \frac{1}{\sqrt{r}}\right) \leq \xi. \tag{27}$$

Based on the two equations and large number theory, we know that the difference between the quantile on samples $q_{1-\epsilon}^r$ and the quantile on the whole dataset $q_{1-\epsilon}$ decays proportionally to $\frac{\tau}{\sqrt{r}}$. In addition, the non-conformity score $s_i$ is defined in Equation (7):

$$s_i = \text{score}(S_i) = |S_i - \mathbb{E}[S]|, \tag{28}$$

where $S_i$ is the number of correct judgments in the $i$-th sample. As the $(1-\epsilon)$-quantile of the sorted scores $s_1 < \ldots < s_r$ at stopping time with $r$ samples is:

$$q_{1-\epsilon}^r = s_{\lceil (1-\epsilon)\cdot(r+1) \rceil}. \tag{29}$$

When the stopping criterion is met, this implies the confidence region for $\mathbb{E}[S]$ has stabilized and the following holds:

$$\mathbb{P}(|S_i - \mathbb{E}[S]| \leq q_{1-\epsilon}^r) = 1 - \epsilon. \tag{30}$$

For the Beta-Binomial mixture model, $\mathbb{E}[S]$ relates to the error rate via:

$$\tilde{P}_{\text{BB}} = \mathbb{P}(S < \lceil k/2 \rceil). \tag{31}$$

We will use the quantile stability argument as follows. For a sequence of independent samples $\{S_1, ..., S_r\}$, let $s_i$ be the non-conformity score defined as:

$$s_i = \text{score}(S_i) = |S_i - \mathbb{E}[S]|, \tag{32}$$

where $S_i$ is the number of correct judgments in the $i$-th sample. By the theory of quantile estimation, for a large enough number of samples $r$, the empirical quantile $q_{1-\epsilon}^r$ converges to the population quantile $q_{1-\epsilon}$ with a known distribution:

$$\sqrt{r}(q_{1-\epsilon}^r - q_{1-\epsilon}) \sim \mathcal{N}\left(0, \frac{\epsilon(1-\epsilon)}{f(q_{1-\epsilon})^2}\right), \tag{33}$$

where $f(\cdot)$ is the density function. This implies:

$$q_{1-\epsilon}^r = q_{1-\epsilon} + O_p\left(\frac{1}{\sqrt{r}}\right), \tag{34}$$

where $O_p(\cdot)$ denotes the order in probability. As the $(1-\epsilon)$-quantile of the sorted scores $s_1 < \ldots < s_r$ at stopping time with $r$ samples is:

$$q_{1-\epsilon}^r = s_{\lceil(1-\epsilon)\cdot(r+1)\rceil}. \tag{35}$$

When the stopping criterion is met, this implies the confidence region for $\mathbb{E}[S]$ has stabilized and the following holds:
$$\mathbb{P}(|S_i - \mathbb{E}[S]| \leq q_{1-\epsilon}^r) = 1 - \epsilon. \tag{36}$$
For the Beta-Binomial mixture model, $\mathbb{E}[S]$ relates to the error rate via:

$$\tilde{P}_{\text{BB}} = \mathbb{P}(S < \lceil k/2 \rceil). \tag{37}$$

By the quantile stability argument above, we have the bound:

$$(1 - \min(\xi, \frac{\tau}{\sqrt{r}}))\mathbb{E}[S]_{\text{BB}} < \mathbb{E}[S]_{\text{adapt}} < (1 + \min(\xi, \frac{\tau}{\sqrt{r}}))\mathbb{E}[S]_{\text{BB}} \tag{38}$$

The error probability of $\tilde{P}_{\text{BB}}$ is defined using the Beta-Binomial cumulative distribution function:

$$\tilde{P}_{\text{BB}} = \mathbb{P}(S < \lceil k/2 \rceil) = F_{\text{BB}}(\lceil k/2 \rceil - 1), \tag{39}$$

where $F_{\text{BB}}$ is the Beta-Binomial cumulative distribution function. Since $F_{\text{BB}}$ is monotonically increasing, the error probability $\tilde{P}_{adapt}$ follows the same proportional bound.

$$(1 - \min(\xi, \frac{\tau}{\sqrt{r}}))\tilde{P}_{\text{BB}} < \tilde{P}_{\text{adapt}} < (1 + \min(\xi, \frac{\tau}{\sqrt{r}}))\tilde{P}_{\text{BB}}. \tag{40}$$

Therefore, we have:

$$\tilde{P}_{\text{adapt}} = (1 \pm \min(\xi, \frac{\tau}{\sqrt{r}}))\tilde{P}_{\text{BB}}. \tag{41}$$

## C  Implementation Details

In this section, we elaborate on the implementation details of `BetaConform`.

We evaluate LLM ensembles with $k \in 1, 3, 5, 7, 9, 11$ models, including GPT-3.5 [Brown et al., 2020], GPT-4 [OpenAI et al., 2024], Llama-3.3-70B [Dubey et al., 2024], Qwen-2.5-72B [Yang et al., 2024], and InternLM-2.5-20B [Cai et al., 2024]. The experiments cover four domains: hallucination detection (HaluEval Li et al., 2023a, TruthfulQA Lin et al., 2021, HalluDial Luo et al., 2024), reasoning (PRM800K Lightman et al., 2023, BIG-bench Srivastava et al., 2022, TRAM Wang and Zhao, 2023), scoring (ICE-Score Zhuo, 2023, Comp-Analysis Zhang et al., 2024), and alignment (JudgeBench Tan et al., 2024, RewardBench Lambert et al., 2024, LLMBar Zeng et al., 2023).

For all experiments, the sampling temperature of LLMs is set to 1, and the random seeds are not fixed. The randomness comes from the $\text{Top} - \text{P}$ sampling of token generation. Each experiment is repeated 30 times to compute the mean and standard deviation of the error margin. The adaptive stopping threshold is set to $\xi = 0.03$ and $\tau = 25$, requiring at least $r \geq 56$ samples to meet the stopping criteria.

## D  Additional Experiments

In Table 4, we conduct ablation studies on our distribution transfer. Compared to ablated variants, our full design achieves the smallest error margin, indicating the effectiveness of our transfer design.

Table 4: The ablation study of `BetaConform` distribution prior transfer. ❶ $\log(r_i) \to r_i$ means the first term $\log(r_i)$ in Eq. 14 is replaced with $r_i$ to still assign a larger dataset higher weight while not considering source datasets could be magnitudes larger. ❷ $\mathrm{CosSim}(\bar{E}_0, \bar{E}_i) \to \frac{1}{|\bar{E}_0 - \bar{E}_i|_2}$ refers to replacing the cosine similarity to measure the source datasets and the target dataset with the reciprocal of the Euclidean distance between the embeddings of the two datasets. This still assigns more similar datasets higher weights. ❸ No $\sigma(\cdot)$ means the transfer weight is computed as $\lambda_i = \log(r_i) \cdot \mathrm{CosSim}(\bar{E}_0, \bar{E}_i)$, without using the sigmoid function $\sigma(\cdot)$ to reduce the weight of low similarity datasets

| Dataset | Ablation | Llama-3.3-70B Error Margin | Qwen-2.5-72B Error Margin | InternLM-20B Error Margin |
|---|---|---|---|---|
| HaluEval | $\log(r_i) \to r_i$ | 10.94 ± 0.57 | 9.53 ± 0.70 | 11.16 ± 0.75 |
| | $\mathrm{CosSim}(\bar{E}_0, \bar{E}_i) \to \frac{1}{|\bar{E}_0 - \bar{E}_i|_2}$ | 11.90 ± 0.85 | 13.17 ± 0.68 | 10.29 ± 0.80 |
| | No $\sigma(\cdot)$ | 10.04 ± 0.23 | 23.03 ± 0.12 | **8.45 ± 0.10** |
| | Ours | **8.82 ± 0.42** | **9.19 ± 0.75** | 8.60 ± 0.64 |
| TruthfulQA | $\log(r_i) \to r_i$ | 13.47 ± 0.66 | 11.17 ± 1.15 | 10.65 ± 0.89 |
| | $\mathrm{CosSim}(\bar{E}_0, \bar{E}_i) \to \frac{1}{|\bar{E}_0 - \bar{E}_i|_2}$ | 15.13 ± 0.71 | 13.14 ± 0.96 | 11.03 ± 0.80 |
| | No $\sigma(\cdot)$ | 6.87 ± 0.01 | 16.52 ± 0.03 | 12.47 ± 0.06 |
| | Ours | **3.37 ± 0.10** | **8.55 ± 0.07** | **10.18 ± 0.10** |
| HalluDial | $\log(r_i) \to r_i$ | 13.55 ± 0.58 | 15.43 ± 0.86 | 10.42 ± 1.00 |
| | $\mathrm{CosSim}(\bar{E}_0, \bar{E}_i) \to \frac{1}{|\bar{E}_0 - \bar{E}_i|_2}$ | 15.54 ± 0.59 | 15.89 ± 0.65 | 10.47 ± 0.67j |
| | No $\sigma(\cdot)$ | **12.39 ± 0.00** | 16.61 ± 0.09 | 13.00 ± 0.07 |
| | Ours | 12.89 ± 0.77 | **13.42 ± 0.53** | **8.72 ± 0.54** |
| JudgeBench | $\log(r_i) \to r_i$ | 25.97 ± 0.03 | 21.23 ± 0.04 | 15.46 ± 0.06 |
| | $\mathrm{CosSim}(\bar{E}_0, \bar{E}_i) \to \frac{1}{|\bar{E}_0 - \bar{E}_i|_2}$ | 14.43 ± 1.12 | 11.26 ± 0.99 | 11.47 ± 0.74 |
| | No $\sigma(\cdot)$ | 24.57 ± 0.44 | 19.26 ± 0.13 | 10.42 ± 0.08 |
| | Ours | **9.45 ± 0.59** | **8.19 ± 0.66** | **8.03 ± 0.54** |
| RewardBench | $\log(r_i) \to r_i$ | 15.00 ± 0.01 | 17.33 ± 0.02 | 20.32 ± 0.01 |
| | $\mathrm{CosSim}(\bar{E}_0, \bar{E}_i) \to \frac{1}{|\bar{E}_0 - \bar{E}_i|_2}$ | 13.29 ± 0.87 | 14.48 ± 0.45 | 16.75 ± 0.34 |
| | No $\sigma(\cdot)$ | 12.88 ± 0.59 | 13.74 ± 0.48 | 16.45 ± 0.26 |
| | Ours | **12.72 ± 0.30** | **12.84 ± 0.48** | **16.35 ± 0.36** |
| LLMBar | $\log(r_i) \to r_i$ | 13.88 ± 0.01 | 15.88 ± 0.01 | 15.45 ± 0.01 |
| | $\mathrm{CosSim}(\bar{E}_0, \bar{E}_i) \to \frac{1}{|\bar{E}_0 - \bar{E}_i|_2}$ | 16.27 ± 0.81 | 15.55 ± 0.83 | 11.90 ± 1.07 |
| | No $\sigma(\cdot)$ | 9.53 ± 0.11 | 13.65 ± 0.01 | 12.58 ± 0.01 |
| | Ours | **8.03 ± 0.39** | **9.95 ± 0.30** | **8.61 ± 0.41** |

# E  Python Implementation

Below we provide the Python-style code for the implementation of our methods

Listing 1: Adaptive Conformal Sampling

```python
import math
import random
import numpy as np

# --- Helper Function for Conformal Sampling ---
def _nonconformity_score_abs_diff_mean(value, mean_value):
    """Calculates L1 distance between a value and the mean as a
        nonconformity measure."""
    return abs(value - mean_value)

# --- Core Function 2: Adaptive Conformal Sampling ---
def run_adaptive_conformal_sampling_for_k_value(
    full_dataset_items,
    k_value_num_models,
    num_samples_per_batch,
```

```python
    max_batches,
    epsilon_conformal=0.05,
    convergence_threshold_q_diff=0.01,
    min_batches_before_stopping_check=5
):
    """
    Performs adaptive sampling for a fixed k-value (number of models)
        using conformal prediction.
    Samples are drawn in batches until the width of the conformal
        interval (related to q-value) stabilizes.

    Args:
        full_dataset_items (list of lists): Each inner list contains
            binary outcomes for a data point across all available
            models.
        k_value_num_models (int): Number of models/outcomes to
            consider from the start of each item.
        num_samples_per_batch (int): Number of items to sample per
            batch.
        max_batches (int): Maximum number of batches to draw.
        epsilon_conformal (float): Significance level for conformal
            prediction (e.g., 0.05 for 95% interval).
        convergence_threshold_q_diff (float): Threshold for q-value
            change to determine stopping.
        min_batches_before_stopping_check (int): Minimum batches
            before checking q-value convergence.

    Returns:
        tuple: (collected_success_counts_S, final_q_value,
            num_batches_processed, sampled_indices_overall)
                - collected_success_counts_S: List of success counts
                    for all sampled items.
                - final_q_value: q-value from conformal prediction at
                    stopping or max batches.
                - num_batches_processed: Actual number of batches
                    processed.
                - sampled_indices_overall: List of original indices of
                    the sampled items.
    """
    if not full_dataset_items:
        return [], None, 0, []
    if not (0 < k_value_num_models <= len(full_dataset_items[0])):
        raise ValueError(f"Invalid k_value_num_models: {
            k_value_num_models}")

    all_collected_S_values = [] # Stores S_i = sum(item[:
        k_value_num_models]) for calibration set
    q_previous = None
    final_q_value = None

    indexed_full_dataset = list(enumerate(full_dataset_items))
    available_indices_for_sampling = list(range(len(
        indexed_full_dataset)))
    sampled_indices_overall = []

    for batch_idx in range(max_batches):
        if len(available_indices_for_sampling) < num_samples_per_batch
            :
            if not available_indices_for_sampling: break # No more
                samples available
            # If remaining samples are less than a batch, sample all
                remaining
            actual_samples_this_batch = len(
                available_indices_for_sampling)
```

```
        else:
            actual_samples_this_batch = num_samples_per_batch

        # Sample indices for the current batch without replacement
            from available indices
        chosen_pool_indices = random.sample(
            available_indices_for_sampling, actual_samples_this_batch)

        current_batch_items = []
        current_batch_original_indices = []

        temp_available_indices = [] # To update available indices for
            the next round

        # Build a set for quick removal of chosen indices
        chosen_pool_indices_set = set(chosen_pool_indices)
        for pool_idx in available_indices_for_sampling:
            if pool_idx in chosen_pool_indices_set:
                original_data_idx, item = indexed_full_dataset[
                    pool_idx]
                current_batch_items.append(item)
                current_batch_original_indices.append(
                    original_data_idx)
            else:
                temp_available_indices.append(pool_idx)
        available_indices_for_sampling = temp_available_indices
        sampled_indices_overall.extend(current_batch_original_indices)

        for item in current_batch_items:
            s_value = sum(item[:k_value_num_models])
            all_collected_S_values.append(s_value)

        if not all_collected_S_values: continue

        s_mean = np.mean(all_collected_S_values)
        nonconformity_scores = [_nonconformity_score_abs_diff_mean(s,
            s_mean) for s in all_collected_S_values]
        nonconformity_scores_sorted = sorted(nonconformity_scores)

        r_calib_size = len(nonconformity_scores_sorted)
        quantile_idx = int(math.ceil((r_calib_size + 1) * (1 -
            epsilon_conformal))) - 1
        quantile_idx = min(max(quantile_idx, 0), r_calib_size - 1) #
            Ensure index is valid

        current_q_value = nonconformity_scores_sorted[quantile_idx]
        final_q_value = current_q_value

        if batch_idx >= min_batches_before_stopping_check -1 : #
            batch_idx is 0-indexed
            if q_previous is not None:
                if abs(current_q_value - q_previous) <
                    convergence_threshold_q_diff:
                    return all_collected_S_values, final_q_value,
                        batch_idx + 1, sampled_indices_overall
            q_previous = current_q_value
        elif batch_idx == 0: # Set q_previous for the first iteration
            q_previous = current_q_value

    return all_collected_S_values, final_q_value, max_batches,
        sampled_indices_overall
```

Listing 2: Mixture of Beta Distributions Fitting via EM

```python
import numpy as np

# --- Helper Function for Distribution Transfer ---
def _normalize_vector(v):
    """L2 normalizes a vector."""
    norm = np.linalg.norm(v)
    return v / norm if norm > 0 else v

# --- Core Function 3: Distribution Transfer for Beta Mixture
#     Parameters ---
def transfer_beta_mixture_parameters(
    target_direct_params,
    source_params_list,
    target_mean_embedding,
    source_mean_embeddings_list,
    target_data_size,
    source_data_sizes_list,
    embedding_similarity_threshold=0.9,
    similarity_scaling_factor=10.0,
    min_source_weight_factor=0.0
):
    """
    Transfers/adjusts Beta mixture parameters from source domains to a
        target domain
    based on embedding similarity and data size.

    Args:
        target_direct_params (tuple): (a1_t, b1_t, a2_t, b2_t, w1_t) -
            Directly estimated parameters for the target domain.
        source_params_list (list of tuples): Each tuple contains
            parameters for a source domain.
        target_mean_embedding (np.array): Mean embedding vector for
            the target domain.
        source_mean_embeddings_list (list of np.array): List of mean
            embedding vectors for source domains.
        target_data_size (int): Number of samples in the target domain
            .
        source_data_sizes_list (list of int): List of data sizes for
            source domains.
        embedding_similarity_threshold (float): Threshold for cosine
            similarity.
        similarity_scaling_factor (float): Scaling factor for the
            similarity score.
        min_source_weight_factor (float): Minimum source weight factor
            , ensuring non-negativity.

    Returns:
        tuple: Transferred parameters (a1_f, b1_f, a2_f, b2_f, w1_f).
    """
    if not source_params_list: # No source, return target's own
        parameters
        return target_direct_params
    if not (len(source_params_list) == len(source_mean_embeddings_list
        ) == len(source_data_sizes_list)):
        raise ValueError("Lengths of source parameters, embeddings,
            and size lists must match.")

    norm_target_emb = _normalize_vector(np.asarray(
        target_mean_embedding, dtype=float))

    weight_target = float(target_data_size)
    source_final_weights = []

    for i in range(len(source_params_list)):
```

```
            norm_source_emb_i = _normalize_vector(np.asarray(
                source_mean_embeddings_list[i], dtype=float))
            similarity = np.dot(norm_target_emb, norm_source_emb_i)

            # Calculate similarity-based weight factor, ensuring non-
                negativity
            similarity_based_factor = similarity_scaling_factor * (
                similarity - embedding_similarity_threshold)
            similarity_based_factor = max(min_source_weight_factor,
                similarity_based_factor)

            current_source_weight = source_data_sizes_list[i] *
                similarity_based_factor
            source_final_weights.append(current_source_weight)

    total_combined_weight = weight_target + sum(source_final_weights)

    if total_combined_weight <= 1e-9: # If total weight is too small,
        return target's own parameters
        return target_direct_params

    num_params_to_transfer = len(target_direct_params)
    final_transferred_params_list = [0.0] * num_params_to_transfer

    # Contribution from target parameters
    for i in range(num_params_to_transfer):
        final_transferred_params_list[i] += weight_target *
            target_direct_params[i]

    # Contribution from source parameters
    for i, src_params_tuple in enumerate(source_params_list):
        if len(src_params_tuple) != num_params_to_transfer:
            raise ValueError(f"Source parameter tuple {i} length
                mismatch with target parameters.")
        for j in range(num_params_to_transfer):
            final_transferred_params_list[j] += source_final_weights[i
                ] * src_params_tuple[j]

    final_params_values = [p / total_combined_weight for p in
        final_transferred_params_list]

    # Post-process parameters: ensure alpha, beta are positive, and w1
        is in [0,1]
    # Assuming the order is (a1, b1, a2, b2, w1)
    a1_f, b1_f, a2_f, b2_f, w1_f = final_params_values

    a1_f = max(a1_f, 1e-6)
    b1_f = max(b1_f, 1e-6)
    a2_f = max(a2_f, 1e-6)
    b2_f = max(b2_f, 1e-6)
    w1_f = np.clip(w1_f, 1e-6, 1.0 - 1e-6)

    return (a1_f, b1_f, a2_f, b2_f, w1_f)
```

Listing 3: Distribution Transfer for Beta Mixture Parameters

```
import math
import random
import numpy as np
from scipy.stats import beta
from scipy.special import betaln, gammaln as lgamma # gammaln is scipy
    's log gamma
from math import comb # math.comb for combinations

# --- Helper Functions for Beta Mixture and Beta-Binomial ---
```

```python
def _replace_elements_for_beta_pdf(probabilities):
    """
    Replaces 0s and 1s in a list of probabilities with close values
    to avoid issues with beta.pdf calculations.
    """
    return [0.999999 if x >= 1.0 else 0.000001 if x <= 0.0 else x for
        x in probabilities]

def _beta_binomial_pmf_log(k_trials, num_successes, alpha, beta_param)
    :
    """
    Calculates the log of the Beta-Binomial PMF: log(P(X=num_successes
        ))
    where X ~ BB(k_trials, alpha, beta_param).
    P(X=x) = C(k,x) * Beta(alpha+x, beta+k-x) / Beta(alpha,beta)
    """
    if not (0 <= num_successes <= k_trials):
        return -np.inf # Log probability of zero

    # Ensure alpha and beta_param are positive
    alpha_stable = max(alpha, 1e-9)
    beta_stable = max(beta_param, 1e-9)

    log_C_k_x = lgamma(k_trials + 1) - (lgamma(num_successes + 1) +
        lgamma(k_trials - num_successes + 1))

    log_beta_num = betaln(alpha_stable + num_successes, beta_stable +
        k_trials - num_successes)
    log_beta_den = betaln(alpha_stable, beta_stable)

    return log_C_k_x + log_beta_num - log_beta_den

def _mixture_beta_binomial_pmf(num_successes, alpha1, beta1, alpha2,
    beta2, w1, k_trials):
    """
    PMF of the mixture Beta-Binomial model:
    P_mix(X=x) = w1 * BB(k, alpha1, beta1) + (1-w1) * BB(k, alpha2,
        beta2)
    """
    log_p1 = _beta_binomial_pmf_log(k_trials, num_successes, alpha1,
        beta1)
    log_p2 = _beta_binomial_pmf_log(k_trials, num_successes, alpha2,
        beta2)

    p1 = np.exp(log_p1)
    p2 = np.exp(log_p2)

    return w1 * p1 + (1 - w1) * p2

# --- Core Function 1: Mixture of Beta Distributions Fitting via EM
    ---
def fit_mixture_of_betas_em(
    raw_samples_outcomes,
    num_trials_per_sample,
    max_iters=100,
    tol=1e-6,
    alpha1_init=None, beta1_init=None,
    alpha2_init=None, beta2_init=None,
    w1_init=None
):
    """
    Fits a mixture of two Beta distributions using the EM algorithm.
    This model is used for modeling observed success rates p_i = (
        successes for sample i) / num_trials_per_sample.
```

```
    Args:
        raw_samples_outcomes (list of lists): Each inner list contains
            binary outcomes (0 or 1) for a data point.
        num_trials_per_sample (int): Number of trials/outcomes to
            consider from the start of each inner list (K or m).
        max_iters (int): Maximum number of iterations for the EM
            algorithm.
        tol (float): Tolerance for convergence.
        alpha1_init, beta1_init, alpha2_init, beta2_init, w1_init:
            Optional initial parameters.

    Returns:
        tuple: (alpha1, beta1, alpha2, beta2, w1) - The estimated
            parameters.
    """
    num_data_points = len(raw_samples_outcomes)
    if num_data_points == 0:
        raise ValueError("Input raw_samples_outcomes cannot be empty."
            )
    if num_trials_per_sample <= 0:
        raise ValueError("num_trials_per_sample must be positive.")

    # Initialize parameters (heuristic based on original code)
    alpha1 = alpha1_init if alpha1_init is not None else 10 *
        num_trials_per_sample
    beta1 = beta1_init if beta1_init is not None else 1 *
        num_trials_per_sample
    alpha2 = alpha2_init if alpha2_init is not None else 1 *
        num_trials_per_sample
    beta2 = beta2_init if beta2_init is not None else 10 *
        num_trials_per_sample
    w1 = w1_init if w1_init is not None else 0.5

    alpha1, beta1 = max(alpha1, 1e-6), max(beta1, 1e-6)
    alpha2, beta2 = max(alpha2, 1e-6), max(beta2, 1e-6)
    w1 = np.clip(w1, 1e-6, 1.0 - 1e-6)

    observed_successes = np.array([sum(sample[:num_trials_per_sample])
        for sample in raw_samples_outcomes])
    proportions = observed_successes / num_trials_per_sample
    proportions_for_pdf = np.array(_replace_elements_for_beta_pdf(
        proportions.tolist()))

    for iteration in range(max_iters):
        # E-Step: Calculate responsibilities
        pdf_vals1 = beta.pdf(proportions_for_pdf, alpha1 + 1e-9, beta1
            + 1e-9) # Add small epsilon for stability
        pdf_vals2 = beta.pdf(proportions_for_pdf, alpha2 + 1e-9, beta2
            + 1e-9)

        numerator1 = w1 * pdf_vals1
        numerator2 = (1 - w1) * pdf_vals2
        denominator = numerator1 + numerator2
        denominator[denominator < 1e-9] = 1e-9 # Avoid division by
            zero

        resp1 = numerator1 / denominator
        resp2 = numerator2 / denominator

        # M-Step: Update parameters (using weighted method of moments
            for Beta parameters)
        w1_new = np.mean(resp1)
        w1_new = np.clip(w1_new, 1e-6, 1.0 - 1e-6)

        # Update alpha, beta for component 1
```

```python
        sum_resp1 = np.sum(resp1)
        if sum_resp1 < 1e-6:
            alpha1_new, beta1_new = alpha1, beta1 # Keep old if weight
                is too small
        else:
            mean_p1_w = np.sum(resp1 * proportions) / sum_resp1
            var_p1_w = np.sum(resp1 * ((proportions - mean_p1_w)**2))
                / sum_resp1
            mean_p1_w = np.clip(mean_p1_w, 1e-6, 1.0 - 1e-6)
            if var_p1_w <= 1e-9 or var_p1_w >= mean_p1_w * (1.0 -
                mean_p1_w) * (1-1e-6): # Check if variance is valid
                # Invalid or too small variance, use heuristic (e.g.,
                    high confidence)
                alpha1_new = mean_p1_w * (num_trials_per_sample * 10)
                    # Larger concentration
                beta1_new = (1.0 - mean_p1_w) * (num_trials_per_sample
                    * 10)
            else:
                common_factor = (mean_p1_w * (1.0 - mean_p1_w) /
                    var_p1_w) - 1.0
                alpha1_new = mean_p1_w * common_factor
                beta1_new = (1.0 - mean_p1_w) * common_factor

        # Update alpha, beta for component 2
        sum_resp2 = np.sum(resp2)
        if sum_resp2 < 1e-6:
            alpha2_new, beta2_new = alpha2, beta2
        else:
            mean_p2_w = np.sum(resp2 * proportions) / sum_resp2
            var_p2_w = np.sum(resp2 * ((proportions - mean_p2_w)**2))
                / sum_resp2
            mean_p2_w = np.clip(mean_p2_w, 1e-6, 1.0 - 1e-6)
            if var_p2_w <= 1e-9 or var_p2_w >= mean_p2_w * (1.0 -
                mean_p2_w) * (1-1e-6):
                alpha2_new = mean_p2_w * (num_trials_per_sample * 10)
                beta2_new = (1.0 - mean_p2_w) * (num_trials_per_sample
                    * 10)
            else:
                common_factor2 = (mean_p2_w * (1.0 - mean_p2_w) /
                    var_p2_w) - 1.0
                alpha2_new = mean_p2_w * common_factor2
                beta2_new = (1.0 - mean_p2_w) * common_factor2

        alpha1_new, beta1_new = max(alpha1_new, 1e-6), max(beta1_new,
            1e-6)
        alpha2_new, beta2_new = max(alpha2_new, 1e-6), max(beta2_new,
            1e-6)

        # Check for convergence
        param_diff = (abs(alpha1 - alpha1_new) + abs(beta1 - beta1_new
            ) +
                    abs(alpha2 - alpha2_new) + abs(beta2 - beta2_new
                        ) +
                    abs(w1 - w1_new))
        if param_diff < tol:
            alpha1, beta1, alpha2, beta2, w1 = alpha1_new, beta1_new,
                alpha2_new, beta2_new, w1_new
            break
        alpha1, beta1, alpha2, beta2, w1 = alpha1_new, beta1_new,
            alpha2_new, beta2_new, w1_new

    return alpha1, beta1, alpha2, beta2, w1

# --- Utility Function: Calculate Majority Vote Success Probability
    from Mixture ---
```

```python
def calculate_majority_vote_success_prob_from_mixture(
    k_trials_for_vote,
    alpha1, beta1,
    alpha2, beta2,
    w1_mixture_weight
):
    """
    Calculates the probability of achieving majority success given
        Beta-Binomial mixture parameters.
    Majority success is defined as number of successes >= ceil(
        k_trials_for_vote / 2).

    Args:
        k_trials_for_vote (int): Total number of trials (e.g., number
            of LLMs).
        alpha1, beta1: Parameters for the first Beta-Binomial
            component.
        alpha2, beta2: Parameters for the second Beta-Binomial
            component.
        w1_mixture_weight (float): Mixture weight for the first
            component.

    Returns:
        float: Probability of majority vote success.
    """
    if k_trials_for_vote <= 0: return 0.0
    majority_threshold = math.ceil(k_trials_for_vote / 2.0)

    prob_sum_for_majority = 0.0
    for num_successes in range(int(majority_threshold),
        k_trials_for_vote + 1):
        prob_sum_for_majority += _mixture_beta_binomial_pmf(
            num_successes, alpha1, beta1, alpha2, beta2,
                w1_mixture_weight, k_trials_for_vote
        )
    return prob_sum_for_majority
```

## F  Limitations and Future Work

The two-component Beta-Binomial mixture improves over simpler models but may still underfit complex judgment distributions. Prior transfer depends on text embedding quality and assumes textual similarity implies similar judgments—an assumption that may not always hold. The current design also focuses on binary/scoring tasks and requires an odd number of annotators.

Future work could explore more flexible mixture models, robust prior transfer methods beyond textual similarity, task-specific features, and extensions to diverse judgment formats and ensemble sizes.

## G  Broader Impacts

BetaConform can reduce the cost of LLM ensemble evaluations, supporting broader use in QA, benchmarking, annotation, and MLOps. It enables scalable, reliable assessment but requires careful attention to estimation error and modeling assumptions, especially in high-stakes applications.

