# OpenReview forum: "$\texttt{BetaConform}$: Efficient MAP Estimation of LLM Ensemble Judgment Performance with Prior Transfer"
_NeurIPS.cc/2025/Conference — NeurIPS 2025 poster_

### Official Review · Reviewer_gw97 · 2025-06-30

**Clarity:** 4
**Significance:** 4
**Originality:** 4
**Rating:** 5
**Confidence:** 3

**Summary:**

This work introduces BetaConform, an economic framework for measuring the accuracy of an ensemble of LLM-based judges. The framework includes (1) a bi-modal beta-binomial mixture model to model the judgment distribution, (2) an adaptive stopping for collecting annotations, and (3) a text-similarity-based transfer mechanism to incorporate mixture parameters learned on other datasets. Extensive experiments show that BetaConform produces higher error margins under the same annotation budget compared to other methods.

**Questions:**

1. On some configurations in Table 2, the prior transfer hurts. Have you found any patterns in the failure cases? For example, it seems like the prior transfer helps Llama-2.2-70B considerably, but slightly hurts GPT-3.5. Why?

**Ethical Concerns:**

["NO or VERY MINOR ethics concerns only"]

**Final Justification:**

The authors propose a principled approach for measuring the performance of an ensemble of LLM-based judges. I'm convinced that the proposed distribution holds in most cases. The early stopping mechanism and transfer mechanism make the method economical to apply. Because of this, I recommend acceptance. This work is especially valuable for the community given the popularity of LLM-based judges and the lack of research into their reliability.

**Limitations:**

Yes

**Paper Formatting Concerns:**

L.255 "Section ??" -> "Section E".

**Quality:**

4

**Strengths And Weaknesses:**

Strengths:
1. The paper is well-written. The motivation, methodology, and experimental results are clear and accessible to readers.
2. The paper tackles an important challenge: measuring the reliability of LLM-based evaluations, particularly in low-data regimes.
3. Extensive experiments and ablations support the design of the framework.

Weaknesses:
1. In the definition of "LLM Ensemble Correct Judgment", Match(·) is binary i.e., a judge is either correct or incorrect, which constrains the types of tasks where this framework is applicable (e.g., ranking candidate responses or multi-label rubrics).
2. The mixture model assumes two peaks (i.e., easy/hard problems), which may not hold on all datasets, in the presence of adversarial prompts, or in sampling regimes with high variability.

---

> ### Author Rebuttal · Authors · 2025-07-30
>
> Thank you, **Reviewer gw97**, for your detailed review and for recognizing our work as economic, well-written, and extensively validated. Below, we address your questions point by point:
>
> **Q1: Binary $\mathrm{Match}(\cdot)$ function**
>
> **A1:** Our rationale for adopting the binary $\mathrm{Match}(\cdot)$ function is that many tasks evaluated by LLM-as-a-Judge can be **naturally framed as binary classification**, resulting in conclusions of "yes" or "no." Additionally, accuracy in scoring-based judgments can be considered binary, where a judgment is **deemed incorrect if the deviation exceeds a certain threshold**, otherwise correct.
>
> We acknowledge that the current approach does not support ranking and appreciate your helpful suggestion. We plan to explore incorporating ranking as part of our future work.
>
> **Q2: Mixture distribution assumption**
>
> **A2:** Our work assumes a typical judging process without adversarial conditions. Adversarial prompts represent special cases of challenging questions designed specifically to **induce incorrect judgments from LLMs**. Our sampling assumption is based on **random availability of samples**.
>
> Although our two-peak assumption originated from observations presented in **Figure 3**, extensive experiments (see **Table 1**) demonstrate that our assumption **consistently yields better performance** compared to baseline methods. This supports the **validity of our two-peak assumption across various datasets**.
>
> We acknowledge the challenge posed by adversarial prompts, which can intentionally create challenging distributions that degrade estimation accuracy. Addressing this will be part of our future research.
>
> **Q3: Performance degradation with GPT-3.5**
>
> **A3:** While distribution transfer aims to enhance estimation accuracy, it may also introduce noise. To mitigate potential negative impacts from the transfer, we designed the transfer mechanism (see **Equation 13**) to carefully balance reliance on source distributions, thus preventing over-dependence and reducing possible degradation.
>
> **Q4: Typos**
>
> **A4:** Thank you for pointing these out. We will revise them in the final manuscript.

---

> > ### Comment · Reviewer_gw97 · 2025-08-05
> >
> > Thanks for the response. I will keep my score.

---

### Official Review · Reviewer_Qt2U · 2025-07-01

**Clarity:** 2
**Significance:** 2
**Originality:** 3
**Rating:** 4
**Confidence:** 3

**Summary:**

The authors propose a problem of estimating the error rate distribution of LLM ensembles. They provide several key points: (1) the distribution of the correct votes can be better modeled by a two-level mixture of Beta-Binomial distribution, compared to the Binomial distribution model; (2) the number of samples can be chosen via the conformal prediction's square-root bounds; and (3) the distribution parameters can be transferred from other source datasets based on text similarities. The algorithm combines the above three points and is verified by empirical evidence.

**Questions:**

1. Line 33's goal should be carefully revised. All the guarantees are made for the population-level rather than arbitrary $x$.

2. About the mixture of the Beta-Binomial model: (1) How to rule out the possibility of the Binomial distribution. Lines 42 and 43 are confusing. Condorcet's Jury Theorem is only correct if (a) each judge has an equal chance of $p>0.5$ making the correct choice; and (b) each judge is making the decision independently. Both conditions are hardly met in the LLM ensemble scenario, especially the second one (I think the LLMs' decisions are highly correlated). So, the Binomial assumption alone doesn't imply that improving the judge number leads to a correct decision with an arbitrarily high chance. The statement is incorrect. (2) How to justify the choice of the mixture of the Beta-Binomial model. I don't quite see the intuition behind this model. The "mixture" step stands for two levels of difficulty. Then what does the "Beta" step stand for? Have you tried other models (e.g., the mixture of Binomials with more mixtures)? I understand that the mixture of Beta-Binomials works the best in your experiments, but the modeling step should be justified more sufficiently.

3. The experiments are using the error margin as the performance metric. As I mentioned in the Weaknesses part, the metric is not sufficient for the entire distribution. Could you provide any further performance metrics to further validate your findings?

4. Typos and minor issues: (1) Line 175 "eta" should be "beta"? (2) Line 237 $\alpha$ and $\beta$ have already been used for Beta distribution's parameters, so please consider replacing the hyperparameters with other characters. (3) Line 255 missing reference. (4) Some texts in the math mode are not in the text mode.

**Ethical Concerns:**

["NO or VERY MINOR ethics concerns only"]

**Final Justification:**

I think the authors have proposed a novel way of modeling the LLM-as-a-judge distribution, and it is worth noticing for the community. But I'm not sure if this distribution is fully justified outside empirical evidence; I'm also not fully convinced of the value of calibrating the marginal error in a dataset rather than the individual one for each prompt. Therefore, I'm giving my score of borderline accept and leaving the decision to the AC.

**Limitations:**

Yes

**Paper Formatting Concerns:**

None.

**Quality:**

3

**Strengths And Weaknesses:**

Strengths:

1. The authors provide a novel problem of estimating the distribution of correct judges when applying LLM ensembles. This may help develop a better understanding of how to model the uncertainty when LLM-as-a-judge is widely adopted today.

2. The paper gives some empirically/theoretically validated key observations (see the Summary part) when modeling the uncertainty: the first is supported by empirical evidence, and the second is supported by conformal prediction theory.

3. The numerical experiments cover a wide range of datasets and models.

Weaknesses:

1. The motivation behind studying this problem is not fully motivated (or at least I do not quite understand). Estimating the total correct votes from an LLM ensemble for an *entire* dataset may be less meaningful compared to estimating the uncertainty on some *certain* prompt (e.g., please see the individual calibration v.s. marginal/populational calibration).

2. The target proposed at the beginning (estimating the whole distribution) is not fully aligned with the numerical experiment metric (estimating the population-level error rate v.s. the true error rate, if I understand correctly). The latter target is *much* easier. To validate the findings, the authors need other metrics (e.g., the estimated $\tau$-quantiles v.s. the actual $\tau$-quantiles for different $\tau$'s). The current status of the experiment doesn't support the goal proposed in line 33.

3. The presentation is a little bit confusing. The paper proposes the target of estimating $\mathbb{P}(S=n|x)$ (line 33), which seems to be an individual/contextual estimation based on observing $x$, yet all the remaining discussions are without the dependence on $x$ (which is population-level). I think the authors need to clarify this point when revising the draft. Besides, the authors need to define what the error rate and the error margin are *formally* in the main text. The authors also need to present the main algorithm framework in the main text rather than in the appendix.

---

> ### Author Rebuttal · Authors · 2025-07-30
>
> Thank you, **Reviewer Qt2U**, for your detailed review and recognizing our work as economic, well-written, and extensively validated. Below, we address your questions point by point:
>
> **W1: Motivation for population-level estimation**
>
> An intuitive example of population-level estimation is a **classification task**, where the model’s accuracy is assessed across the entire dataset rather than a single sample. When given new samples, **the estimated accuracy is the accuracy on the entire dataset**. Following this intuition and standard practice in LLM-as-a-Judge, we adopt population-level performance estimation.
>
> **W2 (1), W3 (1) & Q1: Equation on Line 33**
>
> Thank you for highlighting this. Our intention was to indicate that our task involves estimating the number of correct judgments for any input question, but the original equation missed this assumption. We will revise the equation on **Line 33** as follows:
> $$
> \mathbb{P}(\\#\ \text{correct judgments}=n\ \vert\ \forall x\in D, k\ \text{LLMs judge sample}\ x),
> $$
> where $x$ is a sample and $D$ is the dataset. To achieve this goal, we adopt **population-level estimation to represent the distribution for any given sample**.
>
> **W3 (2): Definition of error rate and error margin**
>
> We apologize for the confusion. Definitions are clarified as follows:
>
> - **Error rate:** The fraction of erroneous judgments made by the ensemble majority vote over the total number of samples in the dataset. It intuitively represents how frequently samples are judged incorrectly.
> - **Error margin:** The absolute difference between the estimated and actual fraction of incorrectly judged samples. This metric assesses the quality of our estimated judgment performance relative to the true performance.
>
> **Q2 (1): Application of Condorcet's Jury**
>
> We reference Condorcet’s Jury Theorem to highlight the inadequacy of assuming a binomial distribution. Given the following reasons, we believe **the Condorcet’s Jury Theorem is applicable**:
>
> 1. $p>0.5$ implies the model performs better than random guessing. This condition **typically holds true** in our experimental settings.
> 2. We respectfully disagree with the notion of high correlation in LLM decisions. While **LLMs judge independently without intercommunication**, decisions are conditionally dependent since judgments are made based on the same question $x$. Thus, the joint judgment distribution $\mathbb{P}(y|x)$ is conditional, not independent.
>
> **Q2 (2): Meaning of “Beta”**
>
> Thank you for this critical inquiry. The motivation behind our choice stems from addressing empirical shortcomings in simpler Binomial models. A Binomial distribution assumes a single, fixed probability of correctness ($p$) across all questions, which contradicts our empirical observations (Figure 3). Introducing the "Beta" component enables modeling the probability itself as a random variable, effectively capturing variations in question difficulty.
>
> **Q2 (3): Further validation of the Beta-Binomial mixture**
>
> To further validate our method, we include additional baseline comparisons involving mixtures of three, four, and five Beta-Binomial distributions, as well as a mixture of two Binomial distributions:
>
> |            | 2 BB Dist. (Ours) |  3 BB Dist.  |  4 BB Dist.  |  5 BB Dist.  | 2 Bin Dist.  |
> | ---------- | :---------------: | :----------: | :----------: | :----------: | :----------: |
> | HaluEval   |    6.68 ± 0.53    | 6.43 ± 0.64  | 7.35 ± 0.51  | 6.73 ± 0.45  | 9.12 ± 0.34  |
> | JudgeBench |    6.98 ± 0.56    | 6.31 ± 0.41  | 6.82 ± 0.43  | 6.98 ± 0.47  | 8.63 ± 0.70  |
> | BIG-bench  |   11.15 ± 0.60    | 12.51 ± 0.53 | 12.35 ± 0.68 | 12.70 ± 0.54 | 12.14 ± 0.24 |
>
> The results show that adding more Beta-Binomial distributions provides **minimal or no performance gain** compared to our two-distribution choice, justifying our decision. Compared to the mixture of Binomial distributions, **our method achieves lower error margins**, suggesting our practice is effective.
>
> **W2 (2) & Q3: Additional metrics for evaluation**
>
> To further assess our method's distribution estimation quality, we provide additional comparisons using **Wasserstein Distance** and **KL Divergence**:
>
> **Table a: Comparison with baselines using Wasserstein Distance**
>
> |           | HaluEval | JudgeBench | BIG-bench |
> | --------- | :------: | :--------: | :-------: |
> | Binomial  |  3.8810  |   3.8404   |  3.1527   |
> | Single BB |  2.0787  |   2.1123   |  1.6547   |
> | Ours      |  1.9614  |   1.7246   |  0.5205   |
>
> **Table b: Comparison with baselines using KL Divergence**
>
> |           | HaluEval | JudgeBench | BIG-bench |
> | --------- | :------: | :--------: | :-------: |
> | Binomial  |  6.2672  |   6.1436   |  4.3358   |
> | Single BB |  3.3084  |   3.1690   |  2.4090   |
> | Ours      |  2.9744  |   2.3681   |  0.3692   |
>
> In the results above, we select HaluEval, JudgeBench and BIG-bench and report the **average numbers across the three datasets**. As shown by the results, our method still outperforms baselines under the **Wasserstein Distance** and **KL Divergence**. This shows that our model is consistently better in different metrics。
>
> **Q4: Typo and minor issues**
>
> Thank you for identifying these issues. We will thoroughly address and correct them in our manuscript revision.

---

> > ### Comment · Reviewer_Qt2U · 2025-08-05
> >
> > Thank you for your detailed reply. I think some major issues have been addressed by the authors, especially the clarity of this paper. Therefore, I'm updating my score to 4. Still, I'm not sure if I have fully understand the statement in the discussion of the Jury Theorem: "We respectfully disagree with the notion of high correlation in LLM decisions. While LLMs judge independently without intercommunication, decisions are conditionally dependent since judgments are made based on the same question $x$. Thus, the joint judgment distribution $\mathbb{P}(y|x)$ is conditional, not independent." Can you explain more to me? Thank you.

---

> > > ### Author Response · Authors · 2025-08-06
> > >
> > > Dear Reviewer Qt2U,
> > >
> > > Thank you for your support and for championing our work’s acceptance!
> > >
> > > We assume Condorcet's Jury Theorem is applicable to the LLM ensemble judgment for the following reasons:
> > >
> > > 1. The single model accuracy is above 50% in most datasets.
> > > 2. LLMs judge independently without any communication.
> > >
> > > Therefore, if the single LLM judge distribution $(k=1)$ follows a  Binomial distribution, Condorcet's Jury Theorem will lead to a decreasing error rate. However, in practice, we do not observe this trend. Given the observation, we propose our mixture of Beta-Binomial distributions.

---

### Official Review · Reviewer_vhVT · 2025-07-02

**Clarity:** 3
**Significance:** 3
**Originality:** 4
**Rating:** 5
**Confidence:** 3

**Summary:**

The paper presents a principled maximum a posteriori (MAP) framework for an economical and precise estimation of the performance of LLM ensemble judgment. It replaces the Binomial assumption with a mixture of Beta-Binomial distributions to capture bimodal judgment patterns, introduces a conformal prediction-driven approach to enable adaptive stopping during iterative sampling, and uses text-similarity-based transfer of learned distributions from source datasets to improve target dataset estimation with few labels.

The BetaConform framework delivers a theoretically guaranteed estimation of LLM ensemble judgment with minimum labeled samples, and also validated empirically.

**Questions:**

Apart from the weaknesses above, I would like to see a discussion on the overall computational overhead, as the cost of text embedding, EM algorithm and conformal prediction may bring extra computational overhead while it effectively reduces annotations.

**Ethical Concerns:**

["NO or VERY MINOR ethics concerns only"]

**Final Justification:**

The paper presents a principled maximum a posteriori (MAP) framework for an economical and precise estimation of the performance of LLM ensemble judgment.  I think the paper may have high practical impacts to the community and therefore recommend accepting the paper.

**Limitations:**

Yes

**Quality:**

4

**Strengths And Weaknesses:**

Strengths:
- The approach is able to dramatically cut down annotation costs.
- The approach is theoretically rigorous in that it is formally grounded in MAP estimation, conformal prediction, and Bayesian transfer.
- The framework is broadly validated on 11 diverse datasets and five SoTA LLMs.

Weaknesses:
- The mixture model only assumes two components (i.e. easy/hard tasks), which may not well generalize to more components that reflect more complicated situations.

- The distribution prior transfer is highly reliant on the quality of text embeddings, which may be a problem when the target dataset has only a few samples.

---

> ### Author Rebuttal · Authors · 2025-07-30
>
> Thank you, **Reviewer vhVT**, for your detailed review and for recognizing our paper as precise, highly efficient, theoretically rigorous, and broadly validated. Below, we address your questions point by point:
>
> **Q1: Two-component mixture distribution**
>
> **A1:** While our initial design of the two-component mixture distribution was based on observations presented in **Figure 3**, we **extensively validated our method across 5 models and 11 datasets**. The results consistently demonstrate that our approach outperforms baseline methods, highlighting its **generalization capability**.
>
> **Q2: Reliance on the quality of text embedding**
>
> **A2:** We argue that, in practice, **ample unlabeled data for computing embeddings is typically available**. Our method significantly reduces the annotation effort required from humans to achieve accurate estimations, making it capable of leveraging the unlabeled data. Moreover, in scenarios where only a few dataset samples are accessible, **our method can directly perform estimations without transfer**, similar to the setting of **Table 1**.
>
> **Q3: Computational overhead**
>
> **A3:** Our method incurs **minimal computational overhead**. The conformal prediction and EM algorithm are executed **solely on a CPU**, typically **completing within three minutes per dataset**. Additionally, adaptive sampling using conformal prediction allows for efficient estimation **using only a small fraction** (approximately 200 samples) of the entire dataset. Furthermore, generating text embeddings relies on efficient model inference, further minimizing computational costs.

---

> > ### Comment · Reviewer_vhVT · 2025-08-09
> >
> > Thanks for clarifying my concerns. I decide to raise my score to 5.

---

### Official Review · Reviewer_NToF · 2025-07-04

**Clarity:** 4
**Significance:** 3
**Originality:** 3
**Rating:** 5
**Confidence:** 3

**Summary:**

The paper proposes a method to efficiently and accurately estimate the judgment performance of LLM ensembles. The method proposed determines: (RQ1): how to estimate the judgment distribution of LLM ensembles and (RQ2): how many samples are needed to achieve a desired estimation under a given error margin threshold. To achieve more accurate estimation, the paper proposes to use a mixture of Beta-Binomial distributions to model the different judgment for "easy" and "hard" questions and use expectation maximization (EM) to estimate the parameters of the distribution from the samples. To achieve the accurate estimation more efficiently, the paper determines the number of samples to use for a good estimation using conformal prediction to compute when to stop sampling when a pre-defined threshold is reached. This limits the number of samples to take while maintaining estimation precision. Finally, to efficiently estimate when only a few samples are available, the paper proposes a text similarity-based method to transfer from the distributions of similar datasets to the target dataset. The paper then conducts experiments across various datasets and shows that the proposed method achieve high estimation precision and achieve low error margin with as few as 10 samples.

**Questions:**

- Why the choice of 2 distributions in the mixture model? Why not 3, 4, or other values? Were other values explored?
- How were the parameter estimates for prior dataset distributions combined with the parameter estimates of the target data? Were the prior parameters used as initial parameters to be further refined through EM? Or were other methods of combining used?
- How will the proposed model change when ensembles with different underlying models are used? Was this explored?

**Ethical Concerns:**

["NO or VERY MINOR ethics concerns only"]

**Limitations:**

Yes

**Quality:**

3

**Strengths And Weaknesses:**

Strengths:
- the paper proposes an original method to efficiently and accurately estimate judgment distribution of LLM ensembles, which combines the choice of distribution (Beta-Binomial) and EM, adaptive stopping (using conformal prediction), and text similarity-based transfer from distributions of existing datasets.
- the problem the paper is tackling is well-motivated, since there is a need to estimate performance of LLM ensembles with few samples. The proposed method can estimate the performance of LLM ensembles with lower human annotation costs (both in terms of time and financial costs).
- while the paper starts from theories in statistics, the proposed method is shown to perform well empirically on multiple datasets.
- the paper is written really well and is very easy to read. The code is given to ensure reproducibility.

Weaknesses:
- while the choice of 2 beta-binomial mixture model is shown to be effective empirically, it is unclear why the authors choose 2 distributions instead of 3, 4, 5, and so on. The reasoning that it is because questions given to LLMs can be categorized into "easy" vs. "hard" seems rather coarse-grained and heuristic. A deeper exploration of this choice will make the paper stronger.
- it will be good to add more intuitive explanations on statistical theories such as the conformal prediction, for readers who are not familiar with the mathematical reasoning behind these theories.
- it is unclear how the distribution transfer was done. How were the parameter estimates for prior dataset distributions combined with the parameter estimates of the target data? Were the prior parameters used as initial parameters to be further refined through EM? Or were other methods of combining used?
- finally, the proposed method seems to be limited in that the LLMs in the ensemble have to all be the same model: same architecture, same size, same version with variations only in terms of the top-P token sampling. This seems to limit the generalizability of the method to real world LLM ensembles that may be composed of different LLM models (e.g., Llama with Qwen).

---

> ### Author Rebuttal · Authors · 2025-07-30
>
> Thank you, **Reviewer NToF**, for the detailed review and acknowledging our paper as original, well-motivated, theory-rooted, empirically strong, and clearly written. Below, we address your questions point by point:
>
> **Q1: The choice of two distributions**
>
> **A1:** We observed that the judge distribution consistently shows **two peaks near all-wrong and all-correct** responses, as illustrated in **Figure 3**. Motivated by this, we adopted a mixture of two Beta-Binomial distributions to **capture this pattern**. Furthermore, our experimental results (**Table 1**) demonstrate that our chosen mixture distribution **outperforms using a single Beta-Binomial distribution**.
>
> To further validate this design choice, we compared our two-distribution mixture with mixtures of three, four, and five distributions, following the setup in **Table 1** on judgments from Llama-3.3-70B:
>
> |            | 2 Distributions (Ours) | 3 Distributions | 4 Distributions | 5 Distributions |
> | ---------- | :--------------------: | :-------------: | :-------------: | :-------------: |
> | HaluEval   |      6.68 ± 0.53       |   6.43 ± 0.64   |   7.35 ± 0.51   |   6.73 ± 0.45   |
> | JudgeBench |      6.98 ± 0.56       |   6.31 ± 0.41   |   6.82 ± 0.43   |   6.98 ± 0.47   |
> | BIG-bench  |      11.15 ± 0.60      |  12.51 ± 0.53   |  12.35 ± 0.68   |  12.70 ± 0.54   |
>
> The results show that adding more distributions provides **minimal or no performance gain** compared to our two-distribution choice, justifying our decision.
>
> **Q2: Clarification of mathematical concepts**
>
> **A2:** Thank you for this valuable suggestion. We will add further explanations in the revised manuscript. Brief, intuitive explanations for key concepts are provided here:
>
> - **Beta-Binomial Distribution and Mixture**: Imagine flipping a coin whose fairness you're unsure about. The Binomial distribution describes how many heads you expect if you know the coin’s bias (probability of heads). The Beta distribution captures your uncertainty about this bias. Combining these, the Beta-Binomial distribution models both the number of heads and your uncertainty about the coin’s fairness. We observed that some questions are "easy" and others are "hard" for LLM judges; thus, we use a **mixture** of two Beta-Binomial distributions to model both question types simultaneously.
> - **Conformal Prediction**: Conformal prediction generates prediction intervals with guaranteed confidence without making strong assumptions about data distributions. It helps determine "how many samples are enough?" by continuously sampling until the prediction interval is acceptably narrow, ensuring efficiency.
> - **Expectation-Maximization (EM)**: EM is an iterative algorithm used to estimate model parameters. It alternates between two steps: the Expectation (E-step), where it estimates missing information based on current parameters, and the Maximization (M-step), where it updates parameters to better fit the data. This process repeats until convergence.
>
> **Q3: Procedure for distribution transfer**
>
> **A3:** We apologize for any confusion. The distribution transfer involves the following steps:
>
> 1. **Estimate distributions** on both target and source datasets using EM, initially **without transfer**.
> 2. **Compute text embeddings** for the target and source datasets (**Lines 230-233**, **Equation 12**).
> 3. **Transfer distributions** from source datasets to the target dataset by refining initial estimates using information from the source datasets (**Lines 234-244**, **Equations 13 and 14**). Specifically, we perform a weighted sum of source and target distributions as described in **Equation 14**.
>
> **Q4: Ensemble of different models**
>
> **A4:** Currently, our method only considers ensembles composed of the same model, aligning with common practice in LLM judging. We plan to explore ensembles of different models as future work.

---

> > ### Comment · Reviewer_NToF · 2025-08-05
> >
> > Thank you for addressing my questions. I keep my score as is.

---

### Author Response · Authors · 2025-08-04

Dear Reviewers,

Thank you again for your dedicated efforts and invaluable feedback on our manuscript.

As the rebuttal deadline is approaching, we are eager to address any further questions you may have. We welcome any additional comments or suggestions for clarification and are prepared to conduct further experiments if needed. We look forward to your guidance to help us improve our paper.

Best regards, \
The Authors

---

### Comment · Area_Chair_rBvE · 2025-08-04
**Reviewer-author discussion phase ends in 2 days**

Dear Reviewers,

Thank you for your reviews for this paper.

This is a kind reminder to please participate in the reviewer-author discussion phase, which ends in 2 days — by Tuesday, August 6 at 11:59 PM AoE. As the authors have responded to your reviews, we kindly ask that you read and engage with their responses, ask clarification questions as needed, and respond to help clarify key points before final decisions.

Your input during this phase is critical to ensuring a constructive and fair outcome.

Let us know if you have any questions or need assistance.

Warm regards,

AC

---

### Note · Authors · 2025-08-15

Dear ACs and Reviewers,

We thank all reviewers for their thoughtful feedback and for recognizing our work as *original*, *well-motivated*, *theory-rooted*, *empirically strong*, *precise*, *efficient*, and *clearly written*. We also appreciate the acknowledgement of our comprehensive evaluation and theoretical grounding.

During the rebuttal, we addressed key concerns across all reviews:

1. **Justification of the Two-Distribution Design** (Reviewers **NToF**, **vhVT**, **Qt2U**, **gw97**): We provided empirical evidence (Tables 1 & new baselines) and comparisons with 3–5 mixture models and binomial mixtures, showing our **two Beta-Binomial mixture** achieves consistently low error margins without overfitting or added complexity.

2. **Clarification of Mathematical Concepts and Assumptions** (Reviewers **NToF**, **Qt2U**, **gw97**): We added intuitive explanations of the Beta-Binomial mixture, conformal prediction, EM, and Condorcet’s Jury Theorem, and clarified definitions for *error rate*, *error margin*, and model independence assumptions.

3. **Distribution Transfer Procedure and Robustness** (Reviewers **NToF**, **vhVT**, **gw97**): We detailed our embedding-based transfer pipeline (Equations 12–14) and explained the transfer weight choice to avoid noise from source datasets, including discussion of scenarios without transfer.

4. **Additional Evaluation Metrics** (Reviewer **Qt2U**): We introduced **Wasserstein Distance** and **KL Divergence** comparisons, confirming our method’s superior distribution estimation quality across multiple datasets.

5. **Computational Efficiency** (Reviewer **vhVT**): We emphasized that our conformal prediction and EM run entirely on CPU within minutes per dataset, requiring only ~200 samples for accurate estimation, thus minimizing annotation and computational cost.

6. **Limitations and Future Directions** (Reviewers **gw97**, **Qt2U**): We acknowledged open directions, including adversarial prompt handling, ranking-based judgments, and heterogeneous model ensembles, as promising extensions of our framework.

We will integrate these clarifications, additional baselines, and extended experiments into the final manuscript to strengthen both empirical and theoretical contributions. We sincerely thank the ACs and reviewers for their constructive feedback, which has greatly improved the clarity, rigor, and completeness of our work.

Best regards, \
Authors

---

### Decision · Program_Chairs · 2025-09-17

**Decision:**

Accept (poster)

**Comment:**

The paper proposes BetaConform, a framework to efficiently and accurately estimate the performance of LLM ensembles. The authors first propose to use binomial judgment to better fit the observed judgment patterns. They then propose to use conformal prediction to enable early stopping during iterative sampling. Finally, they show that the distributional parameters can be transferred to new dataset based on text similarities.

The paper proposes a novel principled approach for efficient judgment estimation from LLM ensembles (NToF, Qt2U), is easy to read (NToF), efficient and effective (vhVT, gw97, Qt2U) and well-motivated (NToF). The theoretical work is supported by comprehensive experiments (vhVT, Qt2U, gw97).

The concerns raised by the reviewers regarding the choice of hyper-parameters (number of components; gw97, vhVT),  generalizability (NToF) and choice of evaluation metrics (Qt2U) were well addressed during the rebuttal. Reviewer Qt2U had concerns about calibrating the marginal error vs errors in individual prompts which was addressed by the author in that they will clarify that they conduct population-level performance estimation and add relevant motivation.

Given the overall clarity, novelty, and strong empirical evidence, as well as the effective rebuttal, I recommend acceptance.